# Surface buoyancy control of millennial-scale variations of the Atlantic meridional ocean circulation

Matteo Willeit[1], Andrey Ganopolski[1], Neil R. Edwards[2], and Stefan Rahmstorf[1]

[1]Potsdam Institute for Climate Impact Research (PIK), Member of the Leibniz Association, P.O. Box 601203, D-14412 Potsdam Germany

[2]Environment, Earth and Ecosystems, The Open University, Walton Hall, Milton Keynes, MK7 6AA, UK

**Correspondence:** Matteo Willeit (willeit@pik-potsdam.de)

**Abstract.**

Dansgaard-Oeschger (DO) events are a pervasive feature of glacial climates. It is widely accepted that the associated changes in climate, which are most pronounced in the North Atlantic region, are caused by abrupt changes in the strength and/or northward extent of the Atlantic meridional overturning circulation (AMOC), possibly originating from spontaneous transitions in the ocean-sea-ice-atmosphere system. Here we use an Earth System Model that produces DO-like events to show that the climate conditions under which millennial-scale AMOC variations occur are controlled by the surface ocean buoyancy flux. In particular, we find that the present day-like convection pattern with deep water formation in the Labrador and Nordic Seas becomes unstable when the buoyancy flux integrated over the northern North Atlantic turns from negative to positive. It is in the proximity of this point that the model produces transitions between different convection patterns associated with strong and weak AMOC states. The buoyancy flux depends on the surface freshwater and heat fluxes and on sea surface temperature through the temperature dependence of the thermal expansion coefficient of seawater. We find that larger ice sheets tend to stabilize convection by decreasing the net freshwater flux while $CO_2$-induced cooling decreases buoyancy loss and destabilizes convection. These results help to explain the conditions under which DO events appear, and are a step towards an improved understanding of the mechanisms of abrupt climate changes.

## 1 Introduction

Most of the Quaternary, except for interglacial states and full glacial states, is characterized by numerous abrupt climate change events (Dansgaard-Oeschger or DO events) (Dansgaard et al., 1993; Rahmstorf, 2002) that are most pronounced in the North Atlantic realm (Hodell et al., 2023; Hoff et al., 2016; Bond et al., 1993) and clearly seen in Greenland ice core data (Andersen et al., 2004), which suggest annual-average air temperature variations of 6–16°C (Kindler et al., 2014). Atmospheric $CO_2$ concentration (Zhang et al., 2021) as well as global ice volume and orbital parameters (Mitsui and Crucifix, 2017; Lohmann and Ditlevsen, 2018) all affect the occurrence and characteristics of DO events. All DO events have a similar time evolution, with an abrupt warming followed by a slow cooling and then a rather abrupt return to stadial (cold) conditions, but their duration varies from several hundred to several thousand years (Lohmann and Ditlevsen, 2019).

It is now generally accepted that the temporal and spatial dynamics of DO events can be explained by abrupt transitions between two modes of the Atlantic Meridional Overturning circulation (or AMOC) (Broecker et al., 1985; Ganopolski and Rahmstorf, 2001), consistent with paleoclimate records (Henry et al., 2016; Bohm et al., 2015; Keigwin and Boyle, 1999; Skinner and Elderfield, 2007). Noise has been argued to be important in triggering the transition between the different AMOC states, either by amplifying the response to a small periodic forcing (Braun et al., 2005; Rahmstorf, 2003) through stochastic resonance (Ganopolski and Rahmstorf, 2002; Alley et al., 2001; Vélez-Belchí et al., 2001), or by exciting and reorganizing internal modes of variability through the coherence resonance mechanism (Timmermann et al., 2003; Pikovsky and Kurths, 1997; Ditlevsen et al., 2005). DO events could therefore originate from internal oscillations within the climate system (Li and Born, 2019; Menviel et al., 2020; Boers et al., 2018), and an increasing number of simplified and general circulation models (GCMs) have revealed spontaneous oscillations resembling DO events with a typical periodicity of 1000 years under a wide range of glacial boundary conditions (Malmierca-Vallet and Sime, 2023). Some GCMs produce internal oscillations for present-day ice sheets but low $CO_2$ (Brown and Galbraith, 2016; Klockmann et al., 2018), others under full glacial conditions (Peltier and Vettoretti, 2014; Romé et al., 2022; Prange et al., 2023) or some combination of mid-glacial (MIS3) conditions (Armstrong et al., 2022; Kuniyoshi et al., 2022; Zhang et al., 2021; Vettoretti et al., 2022) in terms of ice sheets, atmospheric $CO_2$ and orbital parameters.

It is well known that the AMOC is controlled by many factors (e.g. Kuhlbrodt et al., 2007; Nayak et al., 2024), including wind stress, surface buoyancy fluxes and diapycnal mixing. Several studies also suggest the importance of the Southern Ocean and Antarctic bottom water formation in controlling the strength and depth of the AMOC under different climate conditions (e.g. Sun et al., 2020; Oka et al., 2021; Buizert and Schmittner, 2015). However, in this work our main aim is not to explain what controls the strength of the AMOC, but rather focus on what leads to AMOC instability under glacial conditions. Clarifying this is important in order to understand why in reality the DO events occurred under a broad range of glacial climate and boundary conditions, but not during interglacials and peak glacial conditions (Barker et al., 2011; Hodell et al., 2023; Kawamura et al., 2017). It has been suggested that the appearance of DO events is controlled by $CO_2$ (Zhang et al., 2017; Vettoretti et al., 2022), the size of the Northern Hemisphere ice sheets (Zhang et al., 2014; Klockmann et al., 2018; Brown and Galbraith, 2016), and orbital configuration (Zhang et al., 2021). Malmierca-Vallet et al. (2024) recently highlighted the possible role of $CO_2$ concentration in explaining DO variability across different models, independently of the size of the Northern Hemisphere ice sheets and other boundary conditions. However, while the concept of a 'sweet spot' for the occurrence of DO-like variability has recently gained considerable attention, what physical conditions control where it is located in the ice sheet–$CO_2$–orbit space in the different models has remained largely unexplained. Klockmann et al. (2018) and Galbraith and de Lavergne (2019) suggested that the AMOC instability is controlled by the surface density difference between the Southern Ocean and the North Atlantic deep water formation sites, with low $CO_2$, low obliquity and relatively small ice sheets favoring a weak AMOC that is closer to instability. A critical role of Arctic sea ice has also been suggested (Loving and Vallis, 2005) as well as changes in surface winds by glacial ice sheets (Sherriff-Tadano et al., 2021b). Here we use a large number of simulations with an Earth system model to explore the physical control mechanisms behind DO-like variability and propose a key role of the

surface buoyancy flux over the northern North Atlantic in controlling convective instability and the associated abrupt changes in the AMOC.

## 2 Methods

### 2.1 Earth system model

We use the CLIMBER-X (Willeit et al., 2022) Earth system model including a frictional-geostrophic 3D ocean model, a semi-empirical statistical-dynamical atmosphere model, a dynamic-thermodynamic sea ice model and a land surface model with interactive vegetation. All components of the climate model have a horizontal resolution of 5°x5°. CLIMBER-X has a climate sensitivity to a doubling of $CO_2$ of $\sim$3.1°C. The model is described in detail in Willeit et al. (2022) and in general shows performances that are comparable with state-of-the-art CMIP6 models under different forcings and boundary conditions. In particular, the simulated present-day AMOC overturning profile at 26°N in the Atlantic agrees well with observations and is within the range produced by CMIP6 models (Willeit et al., 2022) and the present-day deep convection patterns compare well to ocean reanalysis in the North Atlantic (Fig. 13 in Willeit et al., 2022).

### 2.2 Model experiments

To explore the impact of different aspects of climate on AMOC strength and variability we run an ensemble of climate model simulations using CLIMBER-X with different prescribed ice sheet configurations and atmospheric $CO_2$ concentrations. We performed equilibrium simulations for three different ice sheet configurations, representative of interglacial conditions (present-day, Fig. 1a), full glacial conditions (Last Glacial Maximum GLAC-1D reconstruction (Tarasov et al., 2012), Fig. 1e) and mid-glacial conditions (GLAC-1D reconstruction (Tarasov et al., 2012) for 12 ka, Fig. 1c). The 12 ka GLAC-1D ice sheet reconstruction is similar to the 35 ka ice sheets from PaleoMIST (Gowan et al., 2021), which has been suggested as boundary condition for a DO intercomparison project (Malmierca-Vallet and Sime, 2023), but with a slightly larger Fennoscandian ice sheet. For all three ice sheet configurations we run a set of experiments for 8000 years with prescribed constant atmospheric $CO_2$ concentrations ranging from a pre-industrial value of 280 ppm to 150 ppm, at steps of 10 ppm. The first 3000 years of the simulations are treated as spinup and the following 5000 years are used in the analysis. To isolate the effects of ice sheets and $CO_2$, in all simulations we use present-day orbital parameters.

Since we are not changing the concentration of other greenhouse gases (GHGs) such as $CH_4$ and $N_2O$, these $CO_2$ concentrations should be considered in terms of a $CO_2$ equivalent that implicitly includes the radiative effect of other GHG changes relative to pre- industrial. The $CO_2$ equivalent is roughly $\sim$20 ppm lower than the actual $CO_2$ concentration for most of the last glacial cycle (Appendix A), with minimum values of $\sim$160 ppm during glacial maxima (Fig. A2).

Ice sheets and bedrock topography are prescribed in the simulations, resulting in different surface elevation, bathymetry and land-sea mask for each different ice sheet configuration. The runoff routing directions are automatically derived using a steepest surface gradient approach. In order to conserve water in the climate system, land ice is prevented from melting and

snow accumulating over the ice sheets is cut off at a maximum thickness of 4 m with the excess being routed as 'frozen water' to the ocean, where the latent heat of fusion is also accounted for.

Since the model does not explicitly resolve synoptic-scale and interannual variability in the atmosphere and ocean, to mimic the effect of weather on annual mean AMOC strength we applied perturbations to the surface ocean freshwater flux in the North Atlantic in the latitudinal belt between 50–80°N. We apply a Gaussian white noise with a standard deviation of $0.5\,\mathrm{kg\,m^{-2}\,day^{-1}}$ uniformly over the area and constant over each year. See Appendix B for further details. For mid-glacial ice sheets we also run the simulations with different amplitudes of the noise in the freshwater flux in the North Atlantic, with standard deviations of 0, 0.0625, 0.125, 0.25 and $1\,\mathrm{kg\,m^{-2}\,day^{-1}}$, in addition to the reference simulations with $0.5\,\mathrm{kg\,m^{-2}\,day^{-1}}$. These experiments were performed for the whole range of $CO_2$ concentrations, except for the noise amplitudes of 0.0625 and $0.125\,\mathrm{kg\,m^{-2}\,day^{-1}}$, in which case the simulations were only run for a $CO_2$ concentration of 170 ppm. In all experiments the model is initialized from a 10,000 years long pre-industrial equilibrium spinup with 280 ppm of atmospheric $CO_2$ and present-day ice sheets.

For each ice sheet configuration we additionally performed transient simulations with slowly varying $CO_2$ concentrations: (i) starting at 280 ppm and gradually decreasing $CO_2$ down to 150 ppm and (ii) starting from 150 ppm and gradually increasing $CO_2$ up to 280 ppm. In both cases the rate of change of $CO_2$ is $3\,\mathrm{ppm\,kyr^{-1}}$ implying a total simulation length of ~43,000 years. The initial state for these simulations is an equilibrium with either 280 or 150 ppm of atmospheric $CO_2$. Simulations (i) and (ii) were performed both with no noise in the freshwater flux in the North Atlantic and with the reference noise of $0.5\,\mathrm{kg\,m^{-2}\,day^{-1}}$ and for different diapycnal diffusivities in the upper ocean (Appendix C and Fig. C1).

An additional experiment with freshwater hosing with slowly increasing the freshwater flux from -0.2 to $0.2\,\mathrm{Sv}$ in the North Atlantic (50-70°N) was performed to explore when convective instability is triggered by the addition of freshwater. This experiment was run only without noise and for present-day ice sheets.

## 3  Results

### 3.1  The effect of ice sheets and $CO_2$ on AMOC

For all three ice sheet configurations, when averaged over 5000 years, the AMOC tends to be stronger for higher $CO_2$ concentrations and weakens as $CO_2$ decreases (Fig. 1 and Fig. 2), in line with other models (Brown and Galbraith, 2016; Galbraith and de Lavergne, 2019; Oka et al., 2012; Klockmann et al., 2018; Stouffer and Manabe, 2003; Prange et al., 2023). Moreover, for a given $CO_2$ concentration, the AMOC is generally stronger in experiments with more extensive land ice cover (Fig. 1, Fig. 2 and Fig. 3), again in agreement with other models (Klockmann et al., 2018; Brown and Galbraith, 2016). Therefore, larger ice sheets act to strengthen the AMOC in the model, while lower $CO_2$ concentrations weaken it. This does not contradict the expected AMOC weakening due to anthropogenic $CO_2$ increase, because the latter is an inherently transient phenomenon (Bonan et al., 2022; Stouffer and Manabe, 2003). The net effect of these two counteracting factors leads to a shallower and somewhat weaker AMOC being simulated under full glacial conditions compared to the pre-industrial (Fig. 4i,j). This is in agreement with paleoclimate data (McManus et al., 2004; Bohm et al., 2015; Pöppelmeier et al., 2023), in contrast to most

PMIP models which show a tendency towards a deeper and stronger AMOC at the last glacial maximum compared to the present (Kageyama et al., 2021).

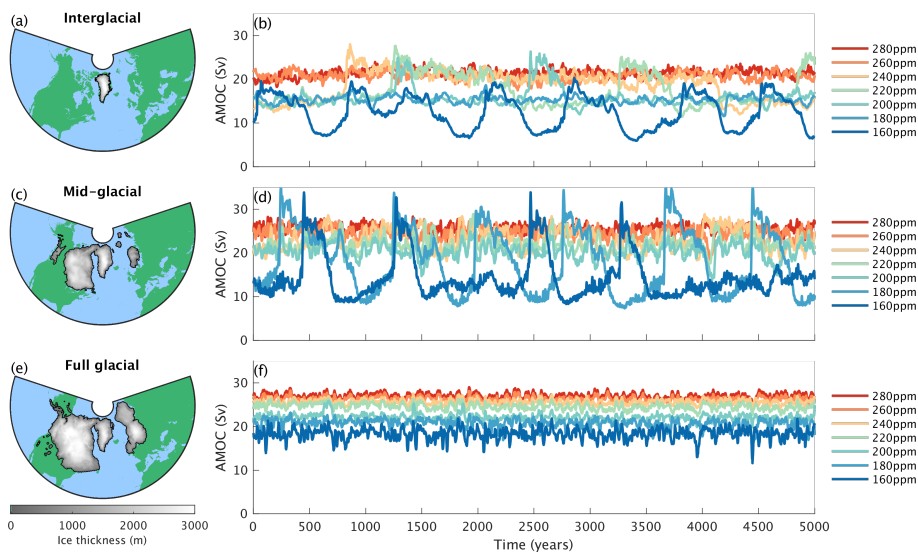

**Figure 1.** AMOC response to ice sheets and CO2. Maximum strength of the Atlantic meridional overturning circulation in model simulations (b,d,f) for the three different ice sheet configurations in (a,c,e) and different prescribed constant equivalent atmospheric $CO_2$ concentrations as shown in the legends on the right.

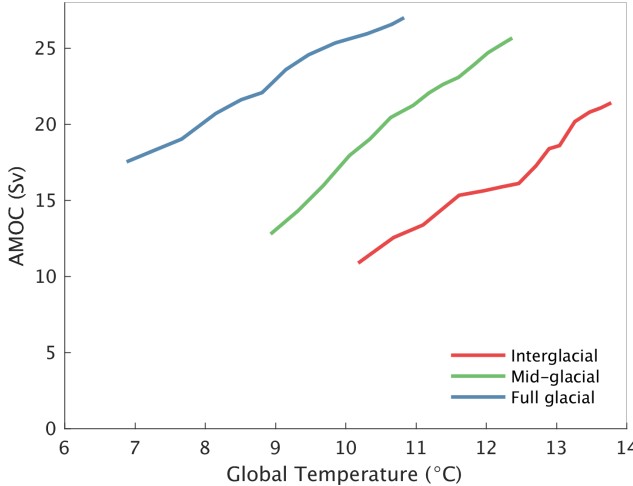

**Figure 2.** AMOC strength as a function of global temperature for the three different ice sheet configurations derived from the ensemble of model simulations with $CO_2$ equivalent ranging between 150 and 280 ppm.

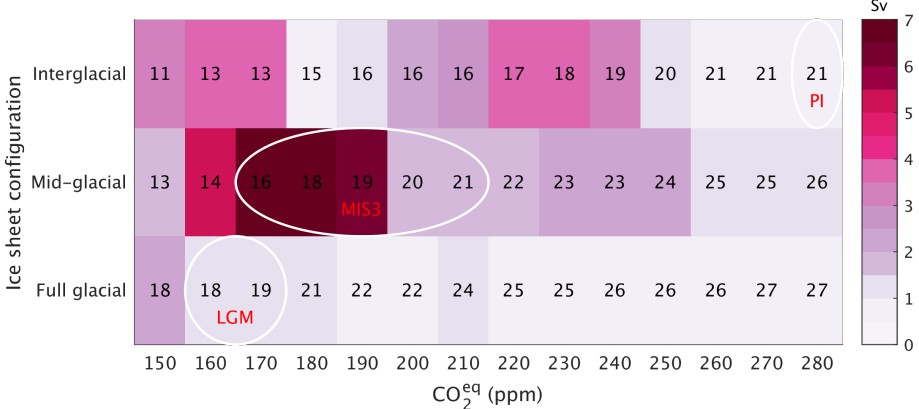

**Figure 3.** AMOC mean and variability. Interannual standard deviation of AMOC time series (shading) and mean AMOC strength (numbers) for different combinations of ice sheet configurations and $CO_2$ concentrations. Conditions representative for the pre-industrial (PI), MIS3 and LGM conditions are indicated.

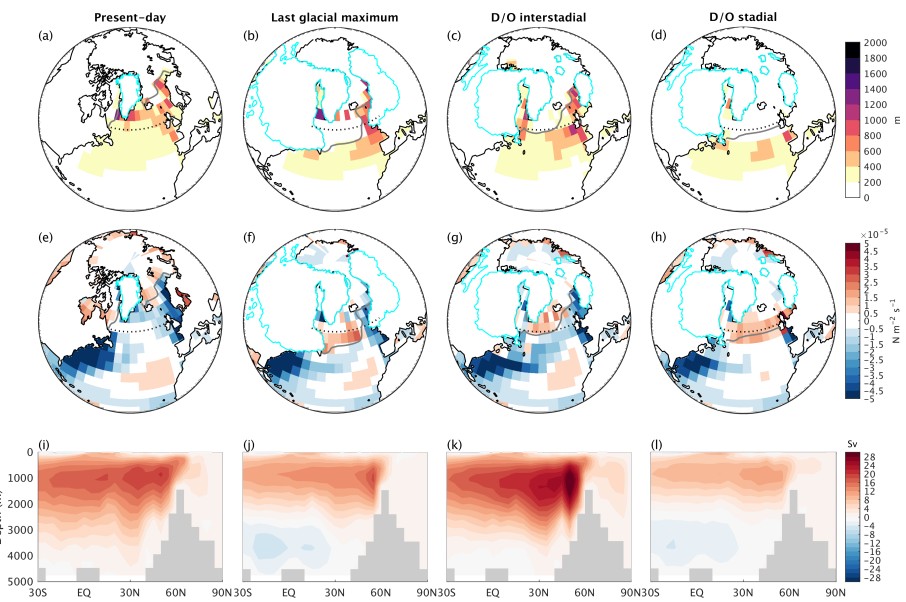

**Figure 4.** Maximum monthly mixed layer depth (a-d), annual mean surface ocean buoyancy flux (e-h) and Atlantic meridional overturning streamfunction (i-l) for different conditions: (left to right) present-day (year 2000 CE) from a transient historical simulation, last glacial maximum (5000 years simulation with LGM boundary conditions following the PMIP protocol (Kageyama et al., 2017) with GLAC-1D ice sheets), interstadial and stadial conditions from the simulation with mid-glacial ice sheets and a $CO_2$ concentration of 180 ppm. The grey line in a-h shows the maximum sea ice extent. The cyan contours indicate the ice sheet extent and the dotted line in panels a-h marks the 55°N latitude.

## 3.2 Internal AMOC variability

Under some combinations of ice sheet configuration and $CO_2$ concentration, the model produces spontaneous oscillations of the AMOC (Fig. 1 and Fig. 3). This is in line with a growing number of models of different complexity producing self-sustained oscillations for specific combinations of ice sheets, $CO_2$ and orbital parameters (Sakai and Peltier, 1997; Schulz et al., 2007; Friedrich et al., 2010; Brown and Galbraith, 2016; Klockmann et al., 2018; Romé et al., 2022; Peltier and Vettoretti, 2014; Zhang et al., 2021; Kuniyoshi et al., 2022; Armstrong et al., 2022). The appearance of internal AMOC oscillations within a
window of $CO_2$ concentrations is fully consistent with recent results from general circulation models (Vettoretti et al., 2022; Malmierca-Vallet et al., 2024).

     The amplitude and period of the oscillations depends on the boundary conditions. Notably, for the mid-glacial ice sheet configuration and $CO_2$ between 150 and 200 ppm, the model produces large amplitude AMOC oscillations that qualitatively resemble DO events recorded in Greenland ice core data both in terms of shape and millennial-scale periodicity (Fig. 1d and
Fig. 5). The sea-saw pattern between Greenland and Antarctic temperatures is also well reproduced, with Antarctic temperature maxima lagging Greenland temperature maxima by $\sim$100 years (Fig. 6b), in agreement with observations (Svensson et al., 2020). The amplitude of the simulated temperature variations over Greenland ($\sim$4°C) is underestimated compared to ice core reconstructions ($\sim$5–15°C), but sea surface temperatures at the Iberian margin vary by $\sim$1.5 °C between stadial and interstadial (Fig. 6c), in very good agreement with proxy records (Martrat et al., 2007). The deficiency in the simulated temperature
response over Greenland in the model is somewhat expected. DO events are expected to affect mainly winter temperature in the northern North Atlantic, primarily as a response to the retreat in sea ice. These temperature changes are going to be largest in a relatively thin layer close to the surface and since in the atmosphere model the transport of heat is mostly horizontal, the warming over the ocean is not very efficiently transported to the summit of the Greenland ice sheet. Also other models, including many GCMs, tend to underestimate the DO warming over Greenland (e.g. Kuniyoshi et al., 2022; Malmierca-Vallet
et al., 2024).

     During the stadial phase deep water is formed south of 55°N (Fig. 4d). This is consistent with the ice core data showing no significant temperature difference over Greenland between stadials with or without Heinrich events, which indicates that the stadial AMOC does not reach far enough north to warm Greenland. The stadial AMOC is also weaker and shallower than at present (Fig. 4i,l), while during the interstadial phase deep convection occurs in the Labrador Sea and the Nordic Seas (Fig. 4c)
resulting in an AMOC that is stronger than at present (Fig. 4k). The northward shift of the deep water formation sites during the interstadial and the associated retreat of sea ice results in annual temperatures up to 15 °C warmer in the Nordic Seas than during stadial conditions (Fig. 7). The simulated change in sea ice cover in the Nordic Seas (Fig. 4c,d and Fig. 7) is in good agreement with proxy-based estimates showing extensive winter sea ice cover over the area during stadials and ice-free conditions during interstadials (Sadatzki et al., 2019, 2020; Hoff et al., 2016; Dokken et al., 2013). A shift from perennial sea
ice during stadials to seasonal sea ice during interstadials is simulated in the Labrador Sea and the Baffin Bay (Fig. 7) and is consistent with reconstructions (Scoto et al., 2022).

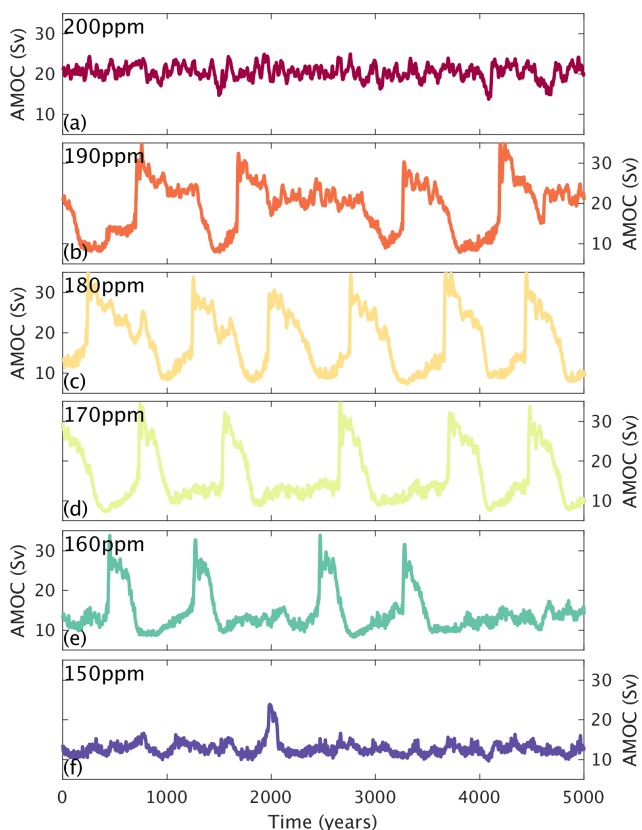

**Figure 5.** Time series of maximum strength of the AMOC streamfunction in simulations with mid-glacial ice sheets and different constant atmospheric $CO_2$ concentrations, decreasing from top to bottom.

For present-day ice sheets, millennial-scale oscillations are simulated for two different $CO_2$ ranges around 230 ppm and 160 ppm (Fig. 1b and Fig. 3). In both cases the AMOC variations are of lower amplitude than for mid-glacial ice sheets. No oscillations are produced by the model for full glacial ice sheets for any of the $CO_2$ concentrations considered (Fig. 1f).

Overall, these results are in qualitative agreement with ice core data showing pronounced millennial-scale climate variability in the North Atlantic during intermediate glacial conditions (e.g. MIS3), but not during peak glacial conditions, such as the LGM, or during interglacials, such as the Holocene.

     The range of boundary conditions under which the oscillations occur depends on the amplitude of the noise that is applied to the surface freshwater flux in the North Atlantic (Fig. 8), with larger amplitude of noise acting to broaden the range. For

mid-glacial ice sheets, self-sustained oscillations are simulated even without noise (Fig. 9). Larger noise levels generally reduce the duration of the stadials, similarly to results obtained using a conceptual coupled climate model (Timmermann et al., 2003). Larger noise levels produce oscillations that are more symmetric and with a shorter period (compare Fig. 9d and f) and could possibly to some extent explain the different characteristics of intrinsic oscillations obtained by Vettoretti et al. (2022) and Kuniyoshi et al. (2022) as opposed to Klockmann et al. (2018). While we find that in our model oscillations can be produced

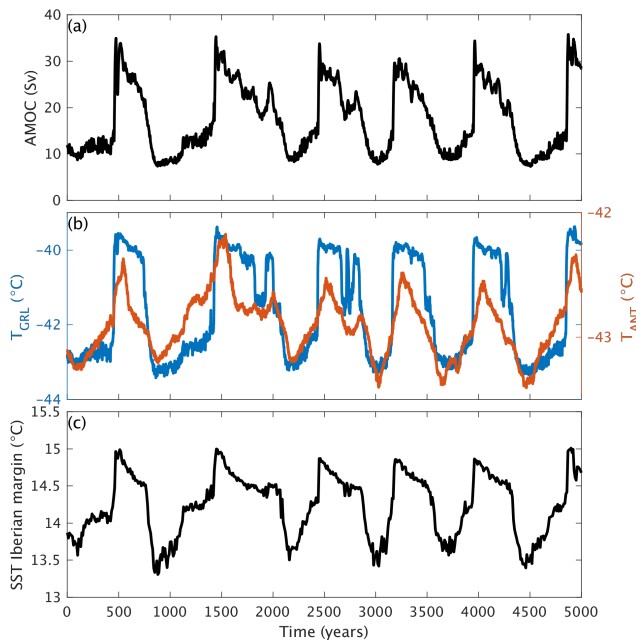

**Figure 6.** (a) Maximum value of the AMOC streamfunction in model simulation with mid-glacial ice sheets and 180 ppm of atmospheric $CO_2$. (b) Corresponding Greenland and Antarctic temperature evolution. Note the different y-axes range. (c) Simulated annual mean sea surface temperature at the Iberian margin.

even without stochastic forcing, noise in the form of interannual variability intrinsic in the climate system could have played an important role in establishing the robust millennial-scale climate variability observed during the Quaternary. Our model has the advantage that it enables a separate investigation of the role of noise on DO dynamics, which can only be partly addressed with GCMs resolving synoptic processes, i.e. by adding additional noise on top of the internally generated variability. However, GCMs cannot remove the noise and can therefore not answer the question of whether noise is crucial for the existence of simulated DO-like events or not.

### 3.3 Convective instability and surface buoyancy flux

Transient simulations with slowly decreasing atmospheric $CO_2$ from 280 to 150 ppm help to elucidate the conditions under which AMOC oscillations occur in the model (Fig. 10). When these experiments are performed with noise in the surface freshwater flux in the North Atlantic, similar behavior is obtained as in the equilibrium simulations shown in Fig. 1. However, performing the same simulations without the imposed noise highlights the presence of discrete transitions in the AMOC at several critical $CO_2$ concentrations, which are specific to each ice sheet configuration. For a range of $CO_2$ values around these points the model shows internal oscillations when noise is applied. Several 'thermal' thresholds in the AMOC have also been recently found by Adloff et al. (2024) in idealized transient model simulations of the last glacial cycle.

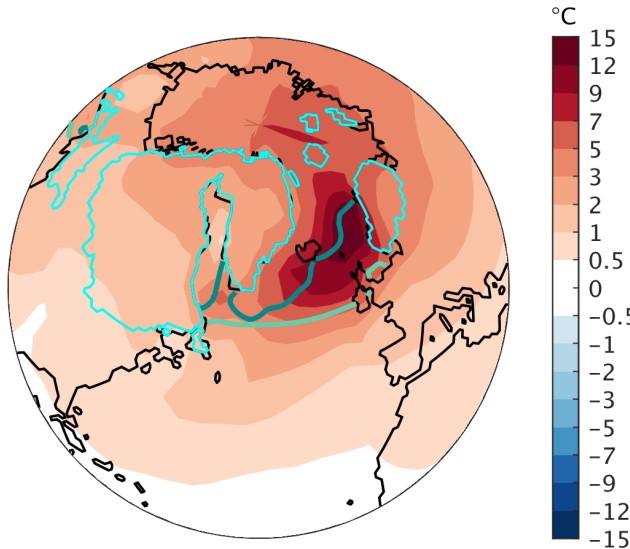

**Figure 7.** Difference in simulated annual mean near-surface air temperature between interstadial and stadial conditions from the experiment with mid-glacial ice sheets and an equivalent atmospheric $CO_2$ concentration of $180\,\mathrm{ppm}$ shown in Fig. 6. The annual mean sea ice extent for the stadial (green) and interstadial (dark-teal) is also shown.

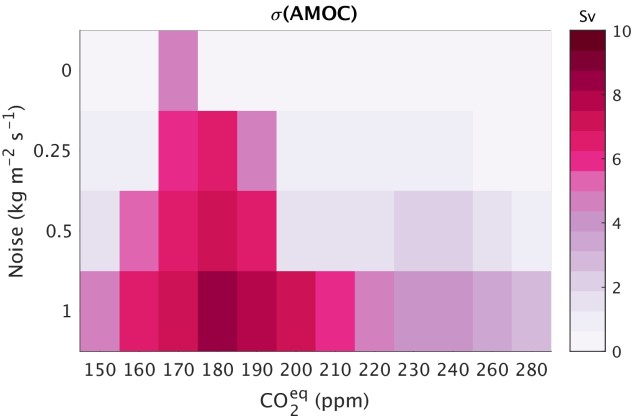

**Figure 8.** Interannual standard deviation of AMOC time series for mid-glacial ice sheets and different combinations of equivalent $CO_2$ concentration and amplitude of the noise applied to the freshwater flux into the North Atlantic.

The different AMOC states are connected to the presence of qualitatively different stable convection patterns in the North
Atlantic for different $CO_2$ values (Fig. 11). Under some conditions, e.g. for present day ice sheets and $CO_2$ between 220 and 240 ppm, two modes of the AMOC corresponding to different convective patterns are stable for the same $CO_2$ (Fig. 10a), but this is not a pre-requisite for the occurrence of internal oscillations.

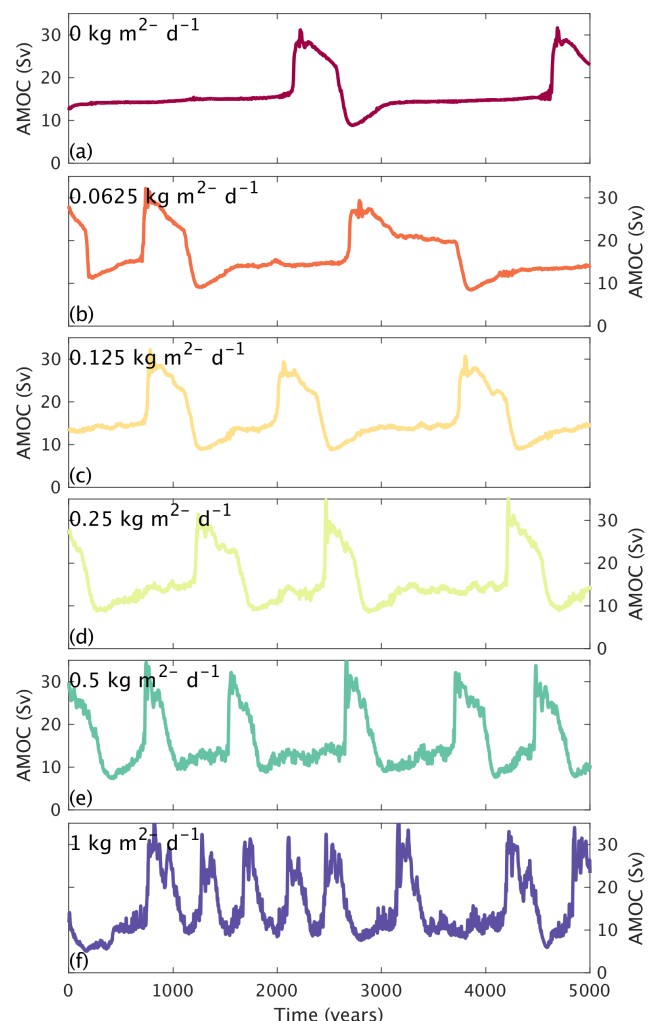

**Figure 9.** AMOC time series for mid-glacial ice sheets, equivalent atmospheric $CO_2$ concentration of $170\,\mathrm{ppm}$ and different amplitudes of the noise applied to the freshwater flux into the North Atlantic, increasing from top to bottom as indicated in the panels.

The stability of the convection patterns, and therefore the AMOC transitions, can be directly linked to the annual mean surface ocean buoyancy flux integrated over the northern North Atlantic and Arctic, $M$. The role of surface buoyancy for AMOC

stability has previously been considered, but rather separately for the different deep water formation regions (Klockmann et al., 2018) and not in terms of an integral value over the whole northern North Atlantic. The use of $M$ to diagnose AMOC instability related to convection processes is based on the following idea. The northern North Atlantic and Arctic regions are characterized by a positive surface freshwater balance as a result of an excess of precipitation over evaporation in combination with freshwater input from rivers. The removal of this freshwater excess from the North Atlantic and Arctic regions can occur

through (i) surface currents transporting low-salinity water to the south, (ii) sea ice export or (iii) deep mixing and evacuation

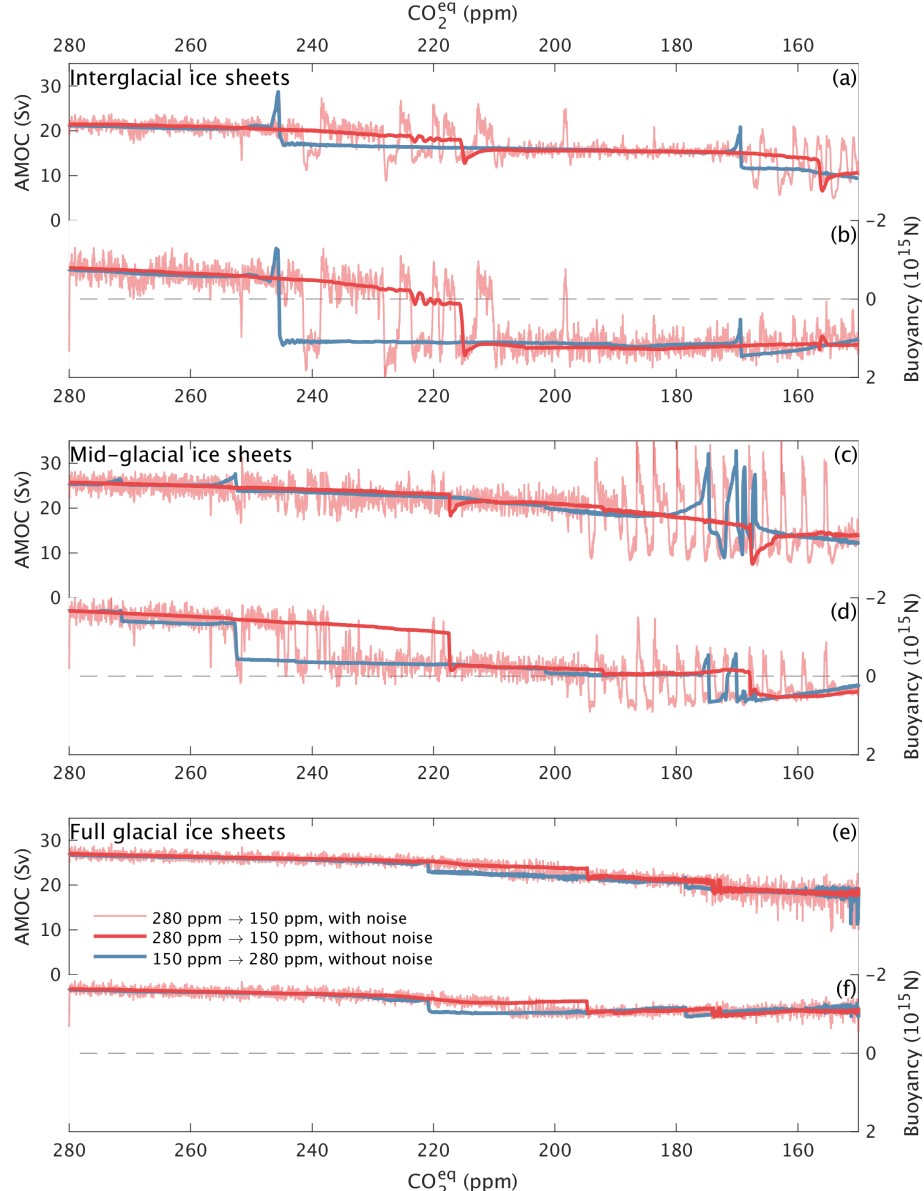

**Figure 10.** Maximum AMOC and integrated surface buoyancy flux, $M$, for experiments with different ice sheet configurations (interglacial, mid-glacial and full glacial, from top to bottom) and slowly varying atmospheric $CO_2$ concentration. The red lines are for simulations with a gradual $CO_2$ decrease starting from pre-industrial conditions with $280\,\mathrm{ppm}$, while the blue line is for a gradual increase of $CO_2$ starting from $150\,\mathrm{ppm}$. The thick lines are experiments without noise in the freshwater flux in the north Atlantic, while the thin line is from an experiment with noise, which is shown only for the $CO_2$ decrease case. The imposed rate of change in $CO_2$ is $3\,\mathrm{ppm\,kyr^{-1}}$ so that the total length of the simulations is $\sim$43,000 years. $M$ is smoothed with a running-mean of 30 years.

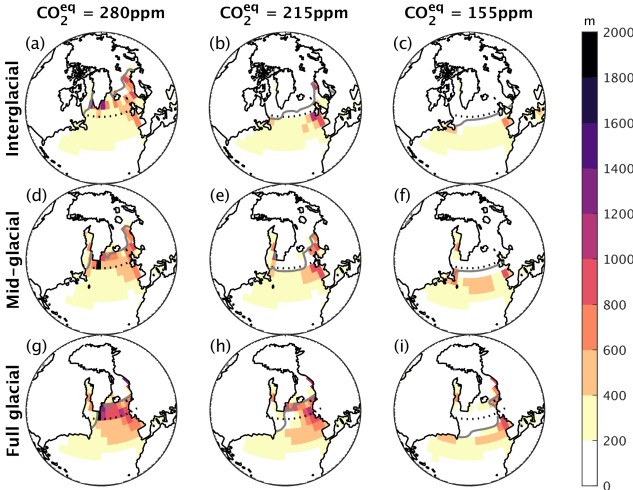

**Figure 11.** Maximum of the monthly mean mixed layer depth in quasi-equilibrium conditions with different ice sheet configurations and $CO_2$ concentrations from the simulations with slowly decreasing $CO_2$ concentration and without noise in the freshwater flux (corresponding to the thick red lines in Fig. 10). The grey lines indicate the maximum sea ice extent.

of the freshwater through the deep ocean. In the case of the interstadial (DO) mode of the AMOC, the (iii) mechanism is the dominant one, while in the stadial mode the mechanisms (i) and (ii) dominate. As shown in Appendix D, the necessary condition for sustaining deep convection is a net negative surface buoyancy flux integrated over the whole northern North Atlantic and Arctic. Our model results confirm that a present day-like spatial organization of convection with deep water forming in the Labrador and Nordic Seas cannot be sustained if $M$ integrated north of $\sim$55°N transitions from negative to positive (Fig. 10b,d). This occurs for $CO_2$ $\sim$220 ppm for interglacial ice sheets and for $CO_2$ $\sim$170 ppm for mid-glacial ice sheets and leads to a sudden shift in the deep-water formation to latitudes south of 55°N (Fig. 11a,b and Fig. 11e,f), with an associated weakening of the AMOC (Fig. 10a,c). The AMOC transition at $CO_2$ $\sim$160 ppm for interglacial ice sheets (Fig. 10a) is not reflected in $M$ (Fig. 10b) because it involves changes in convection pattern that are mostly confined to latitudes south of 55°N. The sudden AMOC weakening at $CO_2$ $\sim$220 ppm for mid-glacial ice sheets (Fig. 10c) involves a reorganization of deep water formation inside the domain north of 55°N and therefore shows a clear imprint on $M$ (Fig. 10d), but does not cause a change of sign of $M$, as convection remains present north of 55°N. The surface buoyancy flux is tightly linked to the convection pattern, with pronounced buoyancy loss concentrated over areas of deep-water formation along the margins of perennial sea ice cover (Fig. 4e-h).

The change in the sign of $M$ can also be seen in the average buoyancy computed for the equilibrium simulations with constant boundary conditions (Fig.12), with a change from negative to positive $M$ around 240 ppm for interglacial ice sheets and around 200 ppm under typical MIS3 conditions.

When noise is applied, the transitions between strong and weak AMOC states are associated with a change in the sign of the buoyancy measure $M$ (Fig. 10b,d). During interstadials, $M$ decreases in magnitude as the heat accumulated in the sub-surface

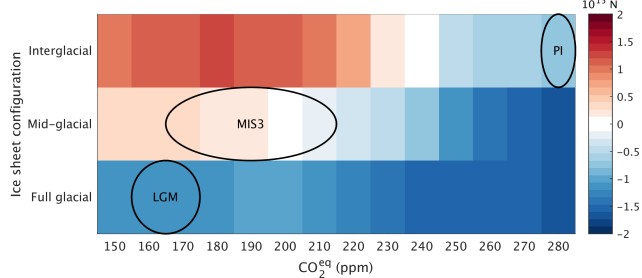

**Figure 12.** Average net buoyancy flux integrated north of $55°$N in the Atlantic ($M$) as a function of atmospheric $CO_2$ concentration for different ice sheet configurations for the simulations with constant boundary conditions shown in Fig.1. The black contours highlight conditions representative for pre-industrial (PI), LGM and MIS3 conditions.

during the stadial conditions is gradually released. The interstadial can only be sustained as long as $M$ is negative, after which there is a rapid transition back to stadial conditions (Fig. 10a-d).

The response of $M$ to decreasing $CO_2$ shown in Fig. 10 results from a combination of factors: (i) the decrease of the thermal expansion coefficient of seawater with decreasing temperatures decreases the thermal surface buoyancy loss, (ii) a weaker AMOC with an associated smaller meridional heat transport decreases the surface sensible heat loss to the atmosphere and therefore also decreases buoyancy loss, while (iii) a decrease in net freshwater flux due to a weakening of the hydrological cycle acts to increase buoyancy loss (Fig. 13 and Fig. 14). The net effect is that the buoyancy loss decreases as $CO_2$ is gradually reduced (Fig. 14d and Fig. 13).

It is noteworthy that for mid-glacial ice sheets $M$ is close to zero over a wide range of $CO_2$ concentrations (Fig. 10d). This is the result of partially compensating effects of the thermal and haline components of the buoyancy flux (Fig. 13) and could help explain the observed ubiquitous appearance of DO events under mid-glacial conditions.

The presence of ice sheets does directly affect the surface buoyancy flux in the northern North Atlantic, with larger ice sheets resulting in increased buoyancy loss and consequently stronger AMOC for any given $CO_2$ concentration (Fig. 10, Fig. 3). In our model, the presence of large Northern Hemisphere ice sheets reduces the net surface ocean freshwater flux into the Atlantic (Fig. 15). Prescribing LGM ice sheets leads to a decrease in the net Atlantic freshwater flux by $\sim 0.1\,\mathrm{Sv}$ compared to experiments with present-day ice sheets, almost independently of the $CO_2$ concentration (Fig. 15), and is a result of the Laurentide ice sheet effectively blocking part of the Pacific-to-Atlantic atmospheric moisture transport in addition to the cooling induced by the presence of the ice sheets which weakens the hydrological cycle over the North Atlantic. Most of the reduction in freshwater flux occurs in the northern North Atlantic, in qualitative agreement with e.g. Eisenman et al. (2009) and Sherriff-Tadano et al. (2021a), thereby increasing the surface buoyancy loss in the deep-water formation regions. CLIMBER-X and PMIP3/4 models show a generally similar $M$ under pre-industrial and LGM conditions (Fig. 16). A decrease in the thermal buoyancy loss at the LGM relative to pre-industrial is by and large compensated by a decrease in the haline buoyancy gain due to a reduced net surface freshwater flux at LGM compared to pre-industrial (Fig. 16). CLIMBER-X tends to show larger haline and thermal buoyancy responses between pre-industrial and LGM compared to most PMIP models. This could be attributable

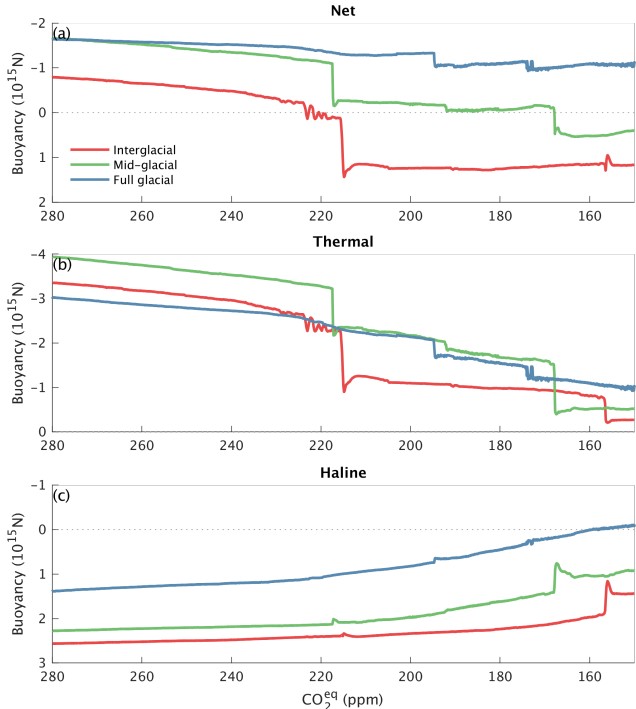

**Figure 13.** (a) Net buoyancy flux integrated north of 55°N in the Atlantic ($M$) as a function of atmospheric $CO_2$ concentration for different ice sheet configurations, with separation into (b) thermal and (c) haline components.

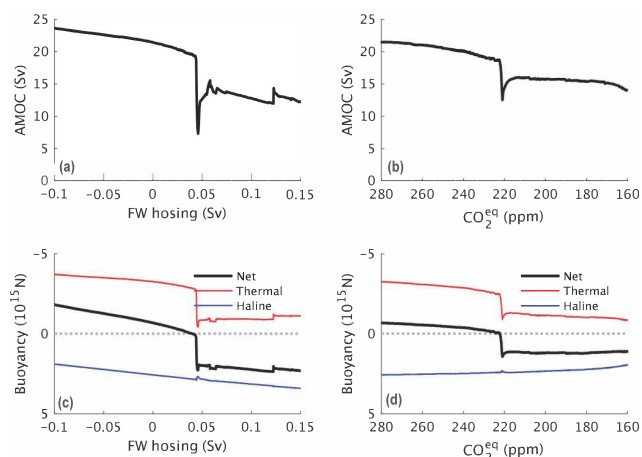

**Figure 14.** (a,b) AMOC strength and (c,d) integrated surface buoyancy flux, $M$, for model simulations with present-day ice sheets and (a,c) slowly increasing freshwater hosing applied to the northern North Atlantic (50–70°N) and (b,d) gradual atmospheric $CO_2$ decrease. In the lower panels $M$ is further separated into thermal and haline components.

to a substantial CLIMBER-X AMOC weakening at LGM as opposed to most PMIP models, which acts to decrease the thermal
buoyancy loss and decrease the haline buoyancy gain due to a cooler northern North Atlantic weakening the hydrological cycle.

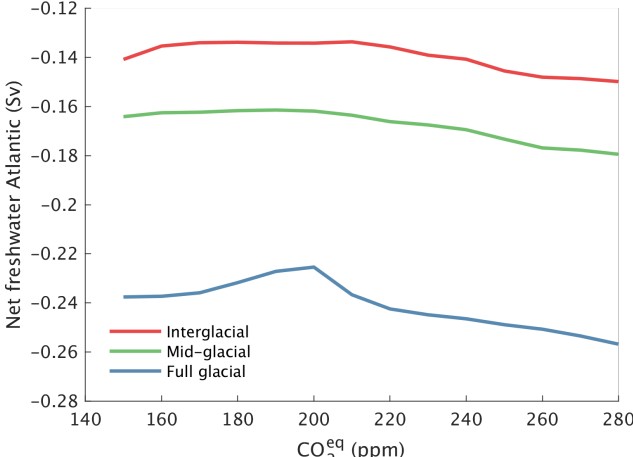

**Figure 15.** Net surface freshwater flux into the Atlantic as a function of atmospheric $CO_2$ concentration for different ice sheet configurations as indicated by the colored lines.

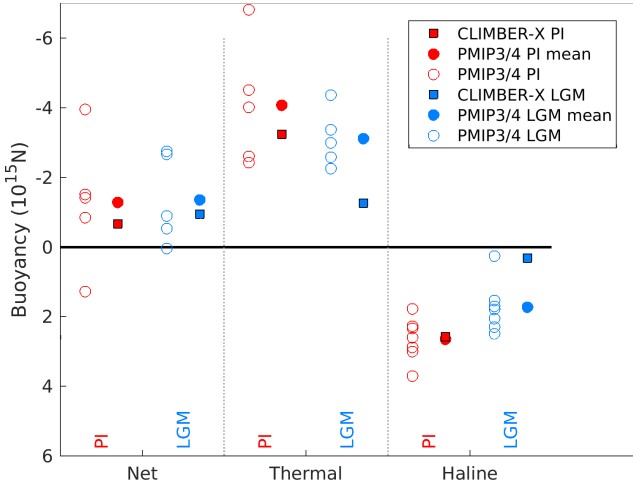

**Figure 16.** Comparison of pre-industrial (PI) and LGM integrated surface buoyancy flux $M$ into the Atlantic north of $55°$N in CLIMBER-X and PMIP3/4 models. The separate contribution of the thermal and haline components of the buoyancy flux are also shown.

Fig. 14 shows that both a slow decrease in atmospheric $CO_2$ and a slow increase in freshwater forcing into the northern North Atlantic produce a gradual decrease in buoyancy loss and eventually trigger an abrupt weakening of the AMOC when $M$ switches from negative to positive. Convective instability can therefore also be triggered by directly perturbing the surface freshwater balance, which affects $M$ (Fig. 14a,c). The noise that is applied to the surface freshwater flux in the model is thus

also directly affecting the surface buoyancy flux and therefore facilitates the transition between different convection states. This also explains why larger noise amplitudes broaden the $CO_2$ range over which oscillations are observed in the model (Fig. 8).

## 3.4 Sensitivity to model parameters

We have also tested the sensitivity of our results to model parameters, specifically the diapycnal diffusivity in the upper ocean (Appendix C and Fig. C1). Extensive work has explored the effect of ocean mixing on AMOC stability, with several studies

showing that larger diapycnal mixing strengthens and stabilizes the AMOC (e.g. Bryan, 1987; Manabe and Stouffer, 1999; Ganopolski and Rahmstorf, 2001; Nof et al., 2007; Prange et al., 2003; Sijp and England, 2006; Schmittner and Weaver, 2001). Peltier and Vettoretti (2014) and Peltier et al. (2020) discussed the role of different diapycnal diffusivities in shaping DO oscillations in their model and Malmierca-Vallet and Sime (2023) note that the different representation of vertical mixing in climate models could explain why some models produce internal DO-like variability under specific boundary conditions,

while others do not. In agreement with previous studies, larger diapycnal diffusivities tend to make the AMOC stronger in CLIMBER-X, therefore also increasing the northward heat transport with a consequent increase in surface sensible heat loss and a decrease in $M$. Consequently, decreasing diapycnal diffusivities leads to an increase in $M$ and brings the system closer to convective instability. This is clearly seen in the simulated response to a slow $CO_2$ decrease, where the critical thresholds for convective instability are systematically shifted to higher $CO_2$ values as diapycnal diffusivity decreases (Fig. 17). Moreover,

smaller diapycnal diffusivities make the internal oscillations more robust, with pronounced oscillations simulated even in the absence of noise in the surface freshwater flux, and also extend the range of $CO_2$ values over which millennial-scale variability is produced by the model (Fig. 17).

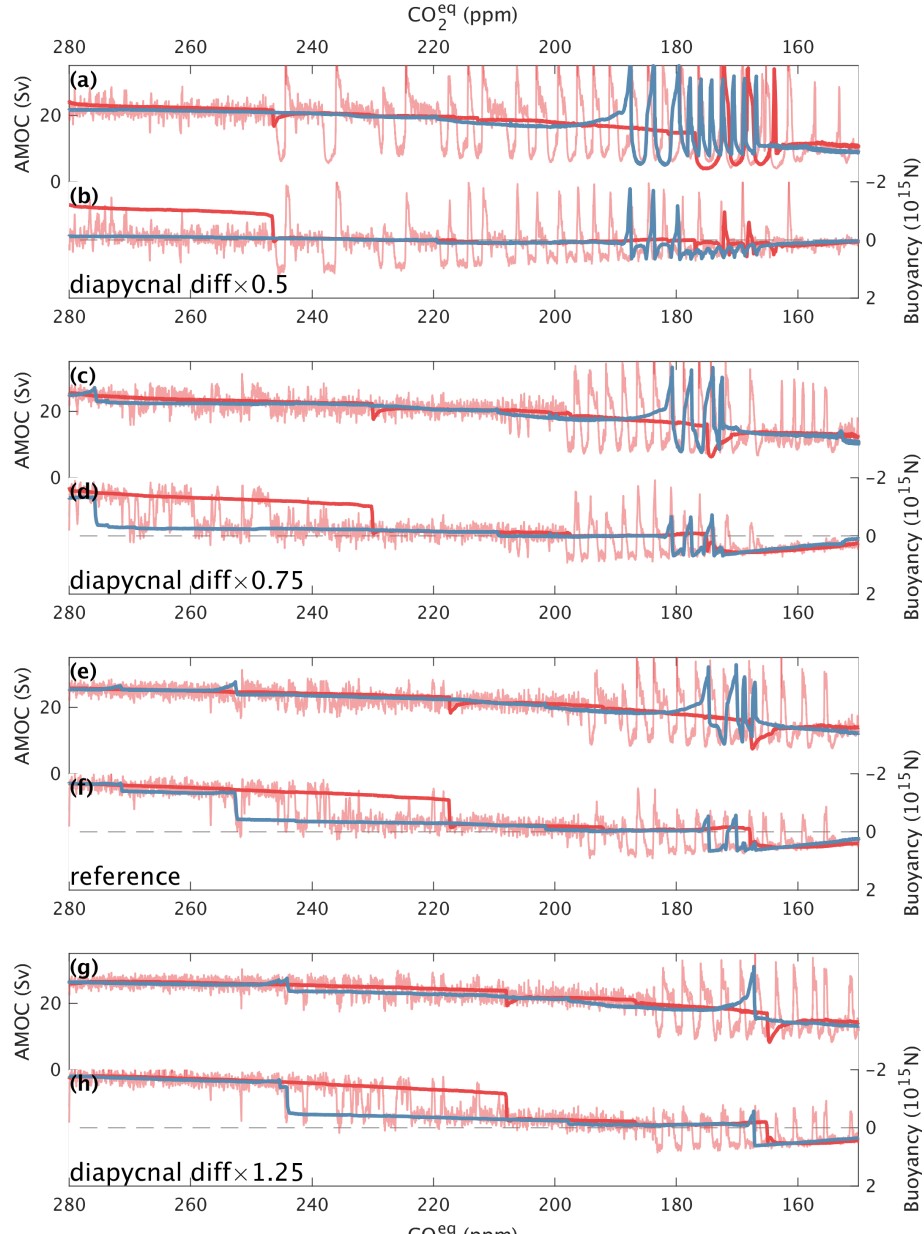

**Figure 17.** Maximum AMOC and integrated surface buoyancy flux, $M$, for experiments with mid-glacial ice sheets and slowly varying atmospheric $CO_2$ concentration for different upper-ocean diapycnal diffusivities (Appendix C). The diffusivities are increasing from top to bottom and the values indicated in the panels specify the scaling factor applied to the value of diapycnal diffusivity at the surface. (e) and (f) are the same as panels (c) and (d) in Fig. 10. The red lines are for simulations with a gradual $CO_2$ decrease starting from pre-industrial conditions with 280 ppm $CO_2$, while the blue line is for a gradual increase of $CO_2$ starting from 150 ppm. The thick lines are experiments without noise in the freshwater flux in the north Atlantic, while the thin line is from an experiment with noise, which is shown only for the $CO_2$ decrease case. The imposed rate of change in $CO_2$ is $3\,\mathrm{ppm\,kyr^{-1}}$ so that the total length of the simulations is $\sim$43,000 years.

## 4 Discussion and conclusions

The stability of the AMOC has historically often been considered in terms of advective instability (Stommel, 1961; Stocker and Wright, 1991; Rahmstorf, 1996; de Vries and Weber, 2005; Hawkins et al., 2011; Hu et al., 2012; Wood et al., 2019), which is controlled by the freshwater budget of the North Atlantic. However, the stability of deep water formation also plays a role in controlling the AMOC (Rahmstorf, 1994, 1995) and in Ganopolski and Rahmstorf (2001) it has been shown that DO events are actually best explained in terms of convective instability. Here we have shown that this convective instability is controlled by the annual mean integrated surface buoyancy flux in the northern North Atlantic ($M$). We also present a theoretical derivation of the integrated buoyancy criterion in Appendix D where we discuss the assumptions and approximations employed to arrive to the buoyancy diagnostic of AMOC instability.

With an *a-posteriori* knowledge of the geographical distribution of deep-water formation sites for different stable convection patterns, it is possible to define a latitude, $\varphi_\mathrm{M}$, which spatially separates the convection sites corresponding to different AMOC modes. The surface buoyancy flux integrated north of $\varphi_\mathrm{M}$ ($M$) is then a measure of whether the convection pattern characterized by deep water formation north of $\varphi_\mathrm{M}$ is stable or not. A negative value of $M$ means that dense surface water is created, which can sustain convection and the formation of deep water north of $\varphi_\mathrm{M}$. Since the net freshwater flux into the northern North Atlantic is positive, the necessary condition for having a negative $M$ is that the surface cools sufficiently through heat loss to the atmosphere.

A change in the freshwater flux into the North Atlantic impacts both advective and convective processes and has different effects on the AMOC depending on the region over which it occurs (Smith and Gregory, 2009; Ganopolski and Rahmstorf, 2001). If applied over deep water formation regions (e.g. 50–70°N) it directly affects $M$ and can therefore trigger convective instability. This is not necessarily the case if the freshwater flux is applied to latitudes further south (e.g. 20–50°N), where it mainly affects the AMOC through its basin-wide effect on the advective salt feedback.

Here we have shown that $M$ close to zero is the condition leading to spontaneous oscillations. $M$ is also expected to be a useful measure for identifying conditions that could lead to the appearance of spontaneous oscillations in complex general circulation climate models. It should be considered as an approximate measure of the stability of convection, with deviations possible if freshwater export through the latitude $\varphi_\mathrm{M}$ by sea ice transport or ocean mixing are important. Ultimately, estimations of $M$ and its relation with convection and AMOC in other models will be needed to assess the robustness of our instability criterion. While the buoyancy criterion is expected to work best in quasi-equilibrium conditions, a further step will be to investigate whether $M$ is also a suitable diagnostic to assess the stability of the AMOC under transient global warming scenarios.

In this paper, the effect of external freshwater forcing on AMOC stability has only been marginally explored, and a comprehensive analysis of AMOC stability in freshwater forcing and $CO_2$ phase space is presented in Willeit and Ganopolski (2024). The changes in $CO_2$ concentration and ice sheet configuration applied in this study also strongly affect the hydrological cycle and thus the net freshwater flux in the Atlantic, but these changes have been taken into account by the model and treated as internal changes. At the same time, it should be noted that ice sheets are prescribed in all of our simulations, whereas in reality

transient changes in ice volume over glacial-interglacial cycles will impact the freshwater balance of the northern North Atlantic and could have a pronounced effect on buoyancy and therefore the conditions favorable for the development of DO-like variability. Transient coupled climate–ice-sheet simulations will be required to address that.

*Code and data availability.* The CLIMBER-X model is freely available as open source code at https://github.com/cxesmc/climber-x. The time series of the model simulations shown in the paper are available from Zenodo: https://doi.org/10.5281/zenodo.8372895. CMIP6 model data are licensed under a Creative Commons Attribution-ShareAlike 4.0 International License (https://creativecommons.org/ licenses) and can be accessed through the ESGF nodes (for instance esgf-data.dkrz.de/search/cmip6-dkrz/). Data from the RAPID AMOC monitoring project is funded by the Natural Environment Research Council and are freely available from www.rapid.ac.uk/rapidmoc.

## Appendix A: Equivalent atmospheric $CO_2$ concentration

The equivalent $CO_2$ concentration is computed taking into account the radiative forcing from $CH_4$ and $N_2O$ gases relative to pre-industrial following Etminan et al. (2016) as described in Willeit et al. (2022) using ice core reconstructions of $CO_2$, $CH_4$ and $N_2O$ from Köhler et al. (2017). The relation between actual and equivalent $CO_2$ concentration and their evolution over the last glacial cycle are shown in Fig.A1 and Fig. A2.

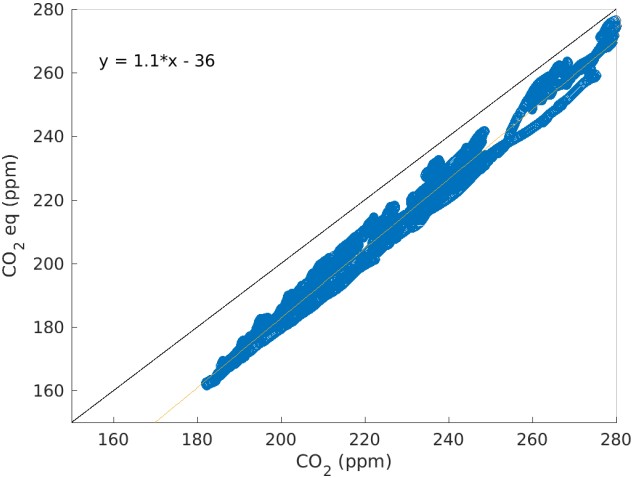

**Figure A1.** Equivalent atmospheric $CO_2$ concentration for radiation versus actual atmospheric $CO_2$ concentration (Köhler et al., 2017) for the last glacial cycle.

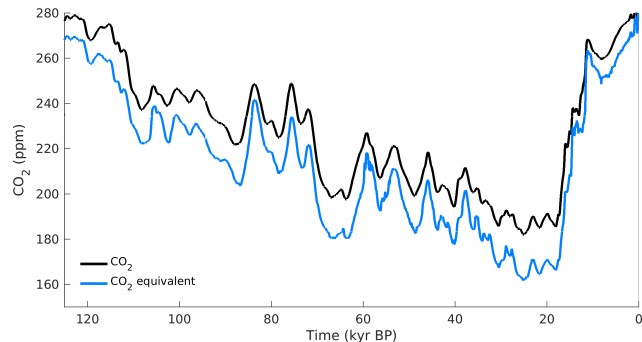

**Figure A2.** Comparison of atmospheric $CO_2$ from ice core data (Köhler et al., 2017) and equivalent $CO_2$ for radiation over the last glacial cycle.

## Appendix B: Noise in the surface freshwater flux

To mimic the effect of weather on interannual AMOC variability we applied perturbations to the surface ocean freshwater flux in the North Atlantic in the latitudinal belt between 50–80°N in the form of Gaussian white noise with a standard deviation of $0.5 \, \mathrm{kg \, m^{-2} \, day^{-1}}$ uniformly over the area and constant over each year. This corresponds to an integrated freshwater flux with a standard deviation of $\sim 0.07 \, \mathrm{Sv}$, which is roughly twice the variability simulated by PMIP3/4 models (Fig. B1). The larger variability used in our study is justified by the fact that the AMOC is affected also by variations in wind stress, sea ice cover and other factors whose interannual variability is not explicitly accounted for in our study. PMIP3/4 models indicate no clear differences between variability at present and at the LGM (Fig. B1), but freshwater variability in the glacial climate could have been larger due to the presence of surrounding ice sheets calving glaciers at irregular intervals. For pre-industrial conditions applying the noise to the freshwater flux results in an interannual AMOC variability with a standard deviation of $\sim 1 \, \mathrm{Sv}$, which is comparable to CMIP6 models (Kelson et al., 2022). Sensitivity tests indicated that the model results are not very sensitive to the details of where the noise is applied, as long as it covers the areas in the North Atlantic where deep water forms.

## Appendix C: Diapycnal diffusivity profiles

Since diapycnal diffusivity is one of the most uncertain parameters in ocean models and often used for model calibration, we designed several different vertical diapycnal diffusivity profiles to test the sensitivity of our results to this important model parameter. In particular, we increased or decreased the diffusivity in the upper ocean by scaling the surface value up or down by factors of 0.5, 0.75 and 1.25. The resulting diapycnal diffusivity profiles are shown in Fig. C1.

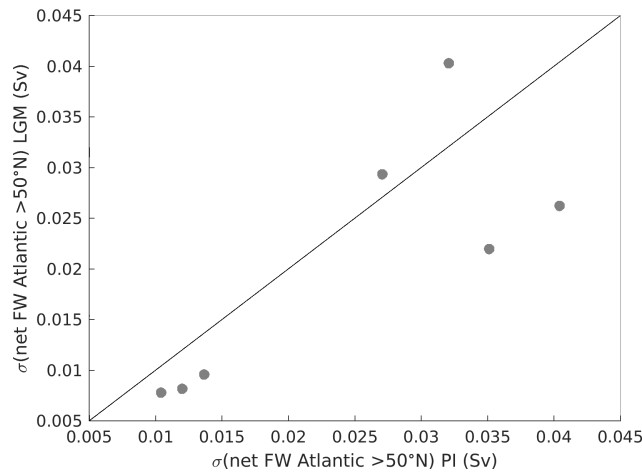

**Figure B1.** Interannual standard deviation of the net freshwater flux into the North Atlantic north of $50°$N as simulated by different PMIP3/4 models at the LGM (y-axis) versus the pre-industrial (PI) (x-axis).

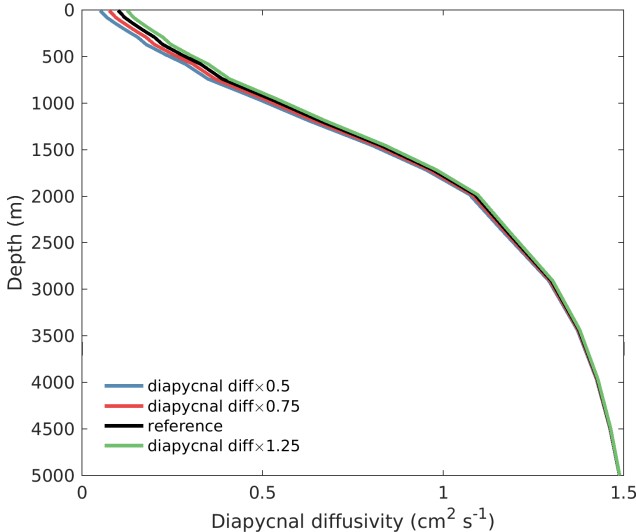

**Figure C1.** Profiles of diapycnal diffusivity used in the parameter sensitivity tests shown in Fig. 17. The legend entries specify the scaling factor applied to the value of diapycnal diffusivity at the surface.

## Appendix D: Theoretical derivation of the integrated surface buoyancy flux criterion

We present a simple 3-cylinder conceptual model (Fig. D1) that aims to explain the convective instability of the AMOC. The purpose of the conceptual model is to prove the following theorem: *The necessary condition for sustaining deep water* 330 *formation in the northern North Atlantic/Arctic (NAA) domain is a negative integrated buoyancy flux over the domain.* The

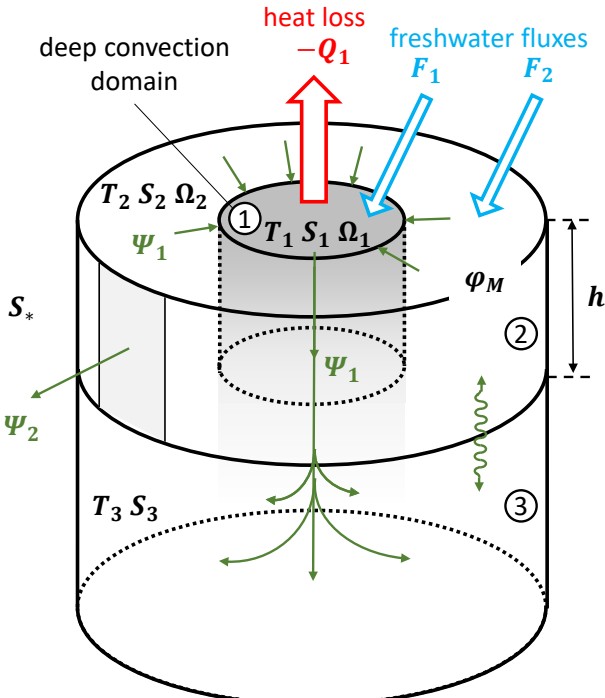

**Figure D1.** Schematic illustration of the idealized northern North Atlantic - Arctic ocean domain used in the theoretical derivation of the integrated surface buoyancy flux criterion.

CLIMBER-X model results presented in the paper further confirm that this is also a sufficient condition. As any conceptual model, the 3-cylinder model is based on a number of simplifications and assumptions. The main assumptions are:

1. Most of the heat transported by the Atlantic ocean through $\varphi_M = 55°N$ is released to the atmosphere in the area of deep water formation (deep convection).

2. Most of the net freshwater flux entering the NAA domain is mixed downward in the deep convection areas and then transported away by the oceanic currents below the surface layer.

3. The system is in (quasi)equilibrium.

An alternative situation that is not described by this model is when deep convection does not operate. In this case, a strong halocline forms over the entire domain and most of freshwater is exported away through the surface layers. Thus, the freshwater balance of the NAA is achieved by different means with and without convection. The conceptual model only aims at determining the criteria for the transition from a convection-on to a convection-off state. It cannot be used to determine the criteria for the resumption of convection, as this requires a model with more boxes.

We consider an idealized NAA ocean domain (Fig. D1) characterized by an area of deep water formation ($\Omega_1$) surrounded by an area $\Omega_2$, which includes all the remaining ocean north of a critical latitude, $\varphi_M = 55°N$. The 3-cylinder model describes

the mean annual temperatures and salinities in volumes 1 and 2 (Fig. D1). The temperature and salinity of the deep ocean (volume 3) are assumed to be constant. Assuming that deep convection has operated during the previous time step, the model determines whether it will operate in the next time step, after a small change in ocean heat transport or freshwater flux or temperature, or a combination of these. If convection operates, then $T_1 \approx T_3$ and $S_1 \approx S_3$, and the equations for temperature and salinity in the upper layer of the convective domain, assuming quasi-equilibrium, can be written in the form:

$$\frac{dT_1}{dt} = \frac{Q_1}{c_p \rho_0 h \Omega_1} + \frac{\Psi_1 (T_2 - T_1)}{h \Omega_1} + K_T \approx 0, \tag{D1}$$

$$\frac{dS_1}{dt} = \frac{1}{h \Omega_1} \left[ -\frac{S_0}{\rho_0} F_1 + \Psi_1 (S_2 - S_1) \right] + K_S \approx 0, \tag{D2}$$

where $Q_1$ is the total surface heat flux into domain 1 (in W), $c_p = 4187\,\mathrm{J\,kg^{-1}\,{}^\circ C^{-1}}$ is the specific heat of water at constant pressure, $\rho_0 = 1000\,\mathrm{kg\,m^{-3}}$ is a reference density, $h \approx 100\,\mathrm{m}$ is a typical thickness of the upper mixed layer and $\Psi_1$ (in $\mathrm{m^3\,s^{-1}}$) is horizontal volume transport from volume 2 to volume 1 and then vertical transport from volume 1 to volume 3, $S_0 = 34.7\,\mathrm{psu}$ is a reference salinity and $F_1$ (in $\mathrm{kg\,s^{-1}}$) is the total surface freshwater flux into domain 1. $K_T$ and $K_S$ represent temperature and salinity fluxes due to convection (vertical mixing), respectively. If there is no convection, these terms are equal to zero. To determine whether convection is operating or not, we follow a similar procedure as used in GCMs with convective adjustment. Equations D1 and D2 are solved in two steps. In the first step, convective fluxes are not applied and these two equations become:

$$\frac{dT_1}{dt} = \frac{Q_1}{c_p \rho_0 h \Omega_1} + \frac{\Psi_1 (T_2 - T_1)}{h \Omega_1}, \tag{D3}$$

$$\frac{dS_1}{dt} = \frac{1}{h \Omega_1} \left[ -\frac{S_0}{\rho_0} F_1 + \Psi_1 (S_2 - S_1) \right]. \tag{D4}$$

In this case $\frac{dT_1}{dt}$ and $\frac{dS_1}{dt}$ may not be negligible. If the new density $\rho_1$, determined by $T_1$ and $S_1$, is greater than $\rho_3$, then convection takes place. But since we assume $\rho_3$ is constant in time, and at the previous time step $\rho_1 = \rho_3$, the condition for convection is that $\frac{d\rho_1}{dt} \geq 0$, which can be expressed as:

$$\frac{d\rho_1}{dt} = -\alpha(T_1) \frac{dT_1}{dt} + \beta \frac{dS_1}{dt} \geq 0, \tag{D5}$$

where $\alpha(T) = 0.052 + 0.012 \cdot T$ is the temperature-dependent thermal expansion coefficient ($\mathrm{kg\,m^{-3}\,{}^\circ C^{-1}}$) and $\beta = 0.8$ is the haline contraction coefficient ($\mathrm{kg\,m^{-3}\,psu^{-1}}$), which is roughly constant. Note that here it is important to consider the non-linearity in the equation of state of seawater and in particular the temperature dependence of the thermal expansion coefficient.

The surface heat loss to the atmosphere over the convective domain $(-Q_1)$ is large and the second term on the right-hand-side (rhs) of eq. D3, representing the lateral advection of ambient temperature between domain 1 and 2, can be neglected. The total surface freshwater flux into domain 1 is given by:

$$F_1 = \int_{\Omega_1} (P - E) \, d\Omega, \tag{D6}$$

where $P$ is precipitation and $E$ is evaporation (both in $\mathrm{kg\,m^{-2}\,s^{-1}}$). Note that there is no contribution from river runoff and sea ice to the surface freshwater balance in the convective domain.

The surface salinity evolution in the rest of the domain is given by:

$$\frac{dS_2}{dt} = \frac{1}{h\Omega_2}\left[-\frac{S_0}{\rho_0}F_2 + \Psi_1\left(S_1 - S_2\right) + \Psi_2\left(S_* - S_2\right)\right] - k\frac{S_2 - S_3}{h^2}. \tag{D7}$$

$S_*$ is the salinity of the upper ocean outside of the considered area, and it is close to $S_0 = 34.7\,\mathrm{psu}$ while $S_2$ is significantly lower. Therefore, it can be assumed that $S_* = S_0$. $F_2$ is the total surface freshwater flux into domain 2 and includes all input from continental runoff ($R$):

$$F_2 = \int_{\Omega_2} \left(P - E + R\right)d\Omega. \tag{D8}$$

As long as there is no export of sea ice out of the whole NAA domain, the freshwater fluxes related to sea ice can be ignored because it forms and melts inside domain 2. The third term on the rhs of eq. D7 represents the 'escape' of freshwater from the NAA domain (e.g. through the southward transport of fresh surface water through the Labrador current). In our model this flux is small (generally <0.01Sv) and can be neglected. The fourth term on the rhs of eq. D7 is the vertical (diffusive) salinity flux ($k$ is the vertical diffusivity in $\mathrm{m^2\,s^{-1}}$), which is also negligible. Using the approximation $dS_2/dt \approx 0$, eq. D7 becomes:

$$\frac{S_0}{\rho_0}F_2 = \Psi_1\left(S_1 - S_2\right), \tag{D9}$$

which, substituted into eq. D4 leads to:

$$\frac{dS_1}{dt} = -\frac{1}{h\Omega_1}\frac{S_0}{\rho_0}\left(F_1 + F_2\right). \tag{D10}$$

Substituting eq. D3 and eq. D10 into eq. D5 then gives:

$$\frac{d\rho_1}{dt} = -\alpha(T_1)\frac{Q_1}{c_p\rho_0 h\Omega_1} - \beta\frac{1}{h\Omega_1}\frac{S_0}{\rho_0}\left(F_1 + F_2\right) \geq 0, \tag{D11}$$

which can be simplified to:

$$\alpha(T_1)\frac{Q_1}{c_p\rho_0} + \beta\frac{S_0}{\rho_0}\left(F_1 + F_2\right) \leq 0. \tag{D12}$$

Given that most of the surface heat loss occurs over the convective region, the total net surface heat flux into the whole domain ($Q$) can be approximated with $Q_1$, while the net surface freshwater flux into the whole domain is $F = F_1 + F_2$. The relation in eq. D12 is then in fact equivalent to the condition that the integral of the surface buoyancy flux over the whole NAA domain ($\Omega$) is negative:

$$\frac{g}{c_p\rho_0}\int_{\Omega}\alpha(T)q\,d\Omega + g\beta\frac{S_0}{\rho_0}\int_{\Omega}f\,d\Omega \leq 0, \tag{D13}$$

where $T$ is the sea surface temperature, $q$ is the net heat flux into the ocean ($\mathrm{W\,m^{-2}}$), $f$ is the net freshwater flux into the ocean ($\mathrm{kg\,m^{-2}\,s^{-1}}$) and $g = 9.81\,\mathrm{m\,s^{-2}}$ is the acceleration due to gravity. Integrating over a full year ($\tau$) we obtain a measure ($M$)

of the buoyancy force (in N) resulting from the application of the buoyancy flux:

$$M = \int_\tau \left[ \frac{g}{c_p \rho_0} \int_\Omega \alpha(T) q d\Omega + g\beta \frac{S_0}{\rho_0} \int_\Omega f d\Omega \right] dt. \tag{D14}$$

The area of integration $\Omega$ for the computation of $M$ is in our case the whole North Atlantic north of $\varphi_M$=55°N, including the Arctic, see Fig D2. The sign of the buoyancy measure $M$ in eq. D14 is what we used in the paper to diagnose convective instability. $M$ for present-day is compared to ocean reanalysis products (Storto and Masina, 2016; Zuo et al., 2019; Haines et al., 2013) and CMIP6 models in Fig. D3.

While the definition of $M$ in eq. D14 makes use of the approximate equation of state of seawater $d\rho = -\alpha(T)dT + \beta dS$, in the model the surface buoyancy flux is computed from the UNESCO equation of state of (Millero and Poisson, 1981), which is used to compute seawater density in CLIMBER-X. However, for the purpose of diagnosing the surface buoyancy flux from observations, reanalysis and models, using a simple quadratic equation of state including the temperature dependence of the thermal expansion coefficient is sufficiently accurate (Roquet et al., 2015).

How effective $M$ is in diagnosing convective instability will depend on the degree to which the assumptions and approximations in the derivation above are met, which could to some extent also be model dependent. Note that if under some circumstances the simplifying assumptions made in the derivation of the criterion in eq. D12 above are not met, a criterion for the stability of the convection can still be derived from the procedure above, but it will simply be a bit more complicated and will not strictly lead to the buoyancy criterion in eq. D14.

As long as the export of sea ice across the critical latitude $\varphi_M$ is negligible, freshwater and latent heat fluxes related to sea ice formation and melt do not have to be accounted for in $q$ and $f$ in the computation of $M$. If the sea ice export is not negligible it should be taken into account by including the related freshwater and heat fluxes over the domain $\Omega$ in the computation of $M$. The $M$ diagnostic in CLIMBER-X is computed using the total net surface freshwater (actually virtual salt flux because of the rigid-lid ocean model) and heat fluxes entering the ocean, which also include sea ice contributions.

It should be noted that the definition of the domain of integration (i.e. $\varphi$) should include the areas of deep water formation and generally the ocean area that can have an effect on the deep convection sites e.g. through freshwater input etc., but it is still arbitrary to some extent and could have an impact on $M$.

*Author contributions.* MW and AG conceived the study. MW designed and performed the model simulations. AG developed the buoyancy framework, with contributions by MW. All authors contributed to the analysis and discussion of the results. MW produced the figures and wrote the manuscript with input from all co-authors.

*Competing interests.* Authors declare that they have no competing interests.

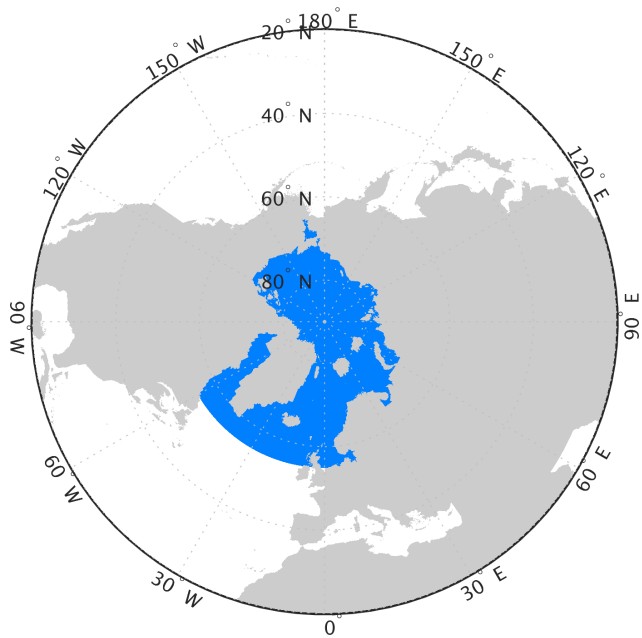

**Figure D2.** Area over which the surface buoyancy flux is integrated to obtain the buoyancy measure $M$, shown here for the mid-glacial boundary conditions case.

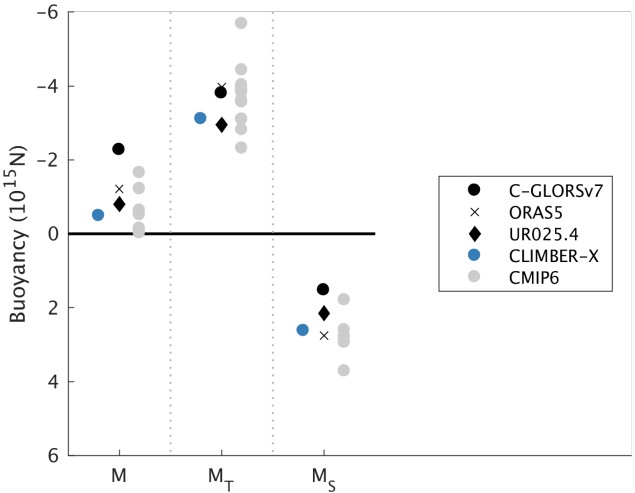

**Figure D3.** Integrated buoyancy flux $M$ for the present day as simulated by CLIMBER-X compared to ocean reanalysis (Storto and Masina, 2016; Zuo et al., 2019; Haines et al., 2013) and CMIP6 models. The thermal ($M_T$) and haline ($M_S$) components are also shown separately.

*Acknowledgements.* We would like to thank Takahito Mitsui for discussions on coherence resonance. We gratefully acknowledge the European Regional Development Fund (ERDF), the German Federal Ministry of Education and Research and the Land Brandenburg for supporting this project by providing resources on the high-performance computer system at the Potsdam Institute for Climate Impact Research. We

acknowledge the World Climate Research Programme, which, through its Working Group on Coupled Modelling, coordinated and promoted CMIP5 and CMIP6. We thank the climate modeling groups for producing and making available their model output, the Earth System Grid Federation (ESGF) for archiving the data and providing access, and the multiple funding agencies who support CMIP5, CMIP6 and ESGF. This research was funded by the German climate modeling project PalMod supported by the German Federal Ministry of Education and Research (BMBF) as a Research for Sustainability initiative (FONA) (grant nos. 01LP1920B, 01LP1917D, 01LP2305B). We would like to thank Sam Sherriff-Tadano and two anonymous reviewers for their valuable comments, which allowed us to substantially improve the manuscript.

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
