# Peer review of "Surface buoyancy control of millennial-scale variations of the Atlantic meridional ocean circulation"

_EGUsphere, 2024_

## Author Comment (AC2)

**Reviewer #1**

*Willeit and colleagues present a large ensemble of CLIMBER-X simulations with various combinations of continental ice sheet configurations and atmospheric CO2 concentrations. This unprecedented ensemble allows them to analyse the physical conditions that determine the forcing range in which CLIMBER-X produces DO-like, millennial-scale climate variability. They find that this "sweet spot" is controlled by the sign of the surface buoyancy flux north of 55N. Millennial-scale transitions between week and strong AMOC states occur when the buoyancy flux north of 55N is about to switch sign. A strong/present day like AMOC occurs when the buoyancy flux is negative and deep water formation takes place in the Labador and Nordic Seas. When the buoyancy flux switches sign, this modern-like deep water formation pattern becomes unsustainable. The conditions under which this sign switch occurs are controlled by the boundary conditions. LGM-like ice sheets tend to enhance buoyancy loss, while low CO2 concentrations tend to decrease it. The balance of the two effects seems to be well captured by CLIMBER-X as the strongest DO-like variability occurs at realistic MIS3-like boundary conditions.*

*Some of the conclusions are not exactly new, e.g. the cancellation of the effects of ice sheet size and CO2 concentration. However the range and combination of covered boundary conditions is unprecedented, and the results are very relevant for the DO- and wider CP community and thus definitely worthy of publication. It is also very much appreciated that the authors define a metric that could be used to compare the physical conditions that control the "sweet spot" across models. Before publication, I would ask the authors to provide more context in some parts and to address a few issues as outlined in my comments below.*

We thank the reviewer for the positive appraisal of our work and the constructive comments.

***Major Comments***

*1. Introduction/Discussion: Please provide more context on what has already been suggested in terms of physical control on the sweet spot. At least Galbraith & de Lavergne (2019) and Klockmann et al (2018) provided some suggestions, e.g. the overall volume of Antarctic Bottom Water (AABW) present in the deep ocean, the density difference between AABW and North Atlantic Deep Water, presence of deep water formation in the Nordic Seas. Also spell out more directly how the additional CLIMBER-X simulations can help in pin-pointing the physical control across models. Because the physical control might also be model dependent.*

We will expand the introduction with a more extensive discussion of what controls AMOC strength in general, and what has been previously suggested in terms of control of the sweet spot in particular. We will also elaborate a bit more on the relevance of our buoyancy criterium and potential limitations that might arise when applying it to other models.

*2. I agree that the buoyancy flux analysis in this paper and the one in Klockmann et al (2018) cannot be compared directly one to one but at least a qualitative comparison should be possible and would actually strengthen the authors arguments even further. This could e.g. take place in the Discussion section.*

*Overall, the mode transitions in the experiments with PI ice sheets in Klockman et al also occur when net buoyancy flux over their NAtl&LabSea region changes from buoyancy loss to buoyancy gain (Klockmann et al use density instead of buoyancy, so the sign is flipped). In*

*their Nordic Seas region, the buoyancy flux is close to zero for the CO2 range where the transition takes place, so the Nordic Seas would not change the sign. This qualitative agreement makes the suggested metric M in the discussion of this manuscript even stronger.*

Calculating buoyancy over the deep water formation area(s), as in Klockman et al. (2018), is of limited use because this flux will be strongly negative as long as deep water formation continues. This is because most of the heat is released in this area, but only a small fraction of the freshwater flux enters the surface through this area (e.g. river runoff along continental margins). Therefore, this flux does not necessarily provide information about the stability of the AMOC. On the contrary, the sign of the buoyancy flux integrated over the entire Atlantic/Arctic ocean domain north of 55N, as shown in our paper, provides useful information about the (convective) stability of the AMOC and explains its instability under glacial conditions. In the revised paper we will add some further discussion on the rationale behind using integrated versus local buoyancy flux in diagnosing AMOC instability.

*One interesting difference can be seen in the effect of ice sheets on the thermal component. In Klockmann et al, the stronger net buoyancy loss with glacial ice sheets is due to increased heat loss over the deep convection sites, while in the present study, it is due to the reduced freshwater input. I do not have an immediate hypothesis where this difference might arise from. Perhaps it is simply due to the different areas of integration.*

This difference could indeed be at least partly due to the different areas of integration. Since most of the heat transported northward by the AMOC will be released over the convection areas, the much stronger AMOC simulated with LGM ice sheets compared to present-day ice sheets (for a given CO2 concentration) will result in a strong increase in the surface sensible heat loss and consequent increase in buoyancy loss over the convection regions, which is well captured by the integration areas in Klockmann et al. 2018. However, the integration areas in Klockmann et al. 2018 capture only part of the changes in hydrological cycle and the resulting changes in the net surface freshwater fluxes.

That said, it is also clear from Fig. 15 that CLIMBER-X tends to show a larger haline buoyancy response between pre-industrial and LGM compared to PMIP models. This could be attributable to a substantial CLIMBER-X AMOC weakening at LGM, which results in a cooler northern North Atlantic and therefore a decrease in precipitation, while most PMIP models, and also the MPI-ESM used in Klockmann et al. 2018, show a strengthening of the AMOC at LGM.

*3. What is the role of sea ice in the buoyancy flux? Is the effect of freezing/brine release and melting included in the freshwater and heat budgets? Sea ice typically plays a big role in feedback loops regarding convection patterns. Even though it can be difficult to determine whether sea-ice is driving the change in the convection patterns or responding to it, it is still worth to be included more explicitly in the analysis.*

The effect of sea ice formation and melt is explicitly included in the computation of the surface buoyancy flux. This will be made more explicit in the revised paper. As long as sea ice is formed and melted inside the area of integration of $M$ (north of 55°N), the net contribution of sea ice to the integrated buoyancy flux will be small and only due to the non-linear equation of state if sea ice is formed and melted in regions which differ in their sea surface temperature.

*4. It might be insightful to show the buoyancy flux also for the equilibrium simulations, e.g. in a similar style as Fig. 3 with buoyancy flux as the colour coding. That would help in linking the results from the transient and equilibrium simulations.*

Thanks for the suggestion, we will consider including the suggested figure in the revised paper.

**Minor Comments**

*l.3 "latitudinal reach" or "northward extent" instead of "latitude reach"?*

We will change it from 'latitude reach' to 'northward extent'.

*l.38-40: see major comment 1*

As outlined in the response to the major comment above, we will expand this section.

*l.45: what is the climate-only setup? Are there other setups?*

CLIMBER-X also includes a global carbon cycle model and an ice sheet model, which are not used in the present study as $CO_2$ and ice sheets are prescribed as constant over time. However, considering that this statement might confuse the readers, we will remove '*in a climate-only setup*'.

*l.75: How sensitive is the model to the area where the noise is applied? Why is it applied only locally and not globally?*

Initial sensitivity tests (not shown in the paper) indicated that the results are not very sensitive to the details of where noise is applied, as long as it covers the areas in the North Atlantic where deep water forms. Noise is introduced in the model to mimic synoptic-scale and interannual climate variability, and applying the same noise globally would be unrealistic as it would assume that this climate variability is globally uniform, which is definitely not the case. A more realistic, global, application of noise would require some kind of weather generator, which is beyond the scope of this study and would very likely have no impact on the results presented in the paper.
In the revised paper we will add the sentence: '*Sensitivity tests indicated that the model results are not very sensitive to the details of where the noise is applied, as long as it covers the areas in the North Atlantic where deep water forms*'.

*l.115: please also state the temperature changes over Greenland in the simulations and in the reconstructions. What does it mean if Greenland change is not capture well but the Iberian margin yes?*

In the revised paper we will explicitly add the modelled and reconstructed temperature changes in Greenland.
The deficiency in the simulated Greenland temperature response in the model is somewhat expected as the atmosphere in CLIMBER-X works best over relatively flat terrain, while the Greenland ice sheet is characterized by large slopes and the circulation over steep slopes is not properly resolved by the model. DO events are expected to affect mainly winter temperature in the northern North Atlantic, primarily as a response to the retreat in sea ice.

This temperature changes are going to be largest in a relatively thin layer close to the surface and since in the atmosphere model the transport of heat is mostly horizontal, the warming over the ocean is not very efficiently transported to the summit of the Greenland ice sheet. Also other models, including many GCMs, tend to underestimate the DO warming over Greenland (e.g. Menviel et al., 2020; Li et al., 2010; Kuniyoshi et al., 2022).

*l.123-124: "The heat transport [...]" What do you base this sentence on? Is it based on previous studies (if yes, please cite)? Or do you infer it from your results (if yes, please elaborate shortly)?*

This sentence is based on our results, but from simulations not shown in the paper. We will therefore delete this sentence in the revised paper.

*Fig.7: Please correct the caption. The interstadial sea-ice extent is drawn in dark teal and not grey*

Will be corrected, thanks.

*Fig.8: In the experiment description and in Fig.9 you mention a total of six noise amplitudes. Here you show only four. Why are 0.0625 and 0.125 not shown? Or did you not cover the full $CO_2$ range for these amplitudes? If so please mention this in the experiment description.*

We performed the 0.0625 and 0.125 noise amplitude simulations only for a $CO_2$ concentration of 170 ppm, which is why those noise levels are not included in Fig. 8. This will be specified in the revised manuscript.

*Fig.9: Which $CO_2$ concentration was used in the respective simulations displayed here?*

The simulations in the figure are for a $CO_2$ concentration of 170 ppm. We will clarify this in the caption.

*l.152: Please briefly state, how do you define stable here (and elsewhere in the manuscript). Also, how realistic are the deep convection patterns in CLIMBER-X given the very coarse resolution?*

Here (and elsewhere), 'stable' will be removed as it does not add any relevant information. The coarse model resolution is obviously a limitation of our model. However, the present-day deep convection patterns compare well to ocean reanalysis in the North Atlantic as shown in Fig. 13 in Willeit et al. 2022. There are unfortunately no reconstructions of the mixed layer depth for DO Stadials and Interstadials, but some information on the convection patterns can be derived from sea ice extent reconstructions, which are tightly linked to the locations of deep water formation. As shown in Fig. 7 and discussed in the text, it seems that the CLIMBER-X sea ice extent change between Stadials and Interstadials is in qualitative agreement with reconstructions, providing some support for the simulated deep water formation patterns.

*l.154: "two modes" instead of "two stable modes". The "stable" in the latter half of the sentence ("are stable under the same CO2") is sufficient.*

Will be fixed, thanks.

*l.161/Fig.10: what about the smaller oscillations that occur around 160ppm with interglacial ice sheets and around 240ppm with mid-glacial ice sheets? In these cases, the buoyancy flux does not change sign.*

The smaller oscillations for interglacial ice sheets and CO2 around 160 ppm are not reflected in the buoyancy flux because they involve changes in convection pattern that are mostly confined to latitudes south of 55°N, which is therefore not reflected in $M$.
The oscillations at ~240 ppm for mid-glacial ice sheets involve a reorganization of deep water formation inside the domain north of 55°N. This therefore shows a clear imprint on $M$, but does not cause a change of sign of $M$, as convection remains present north of 55°N.
We will add this discussion in the revised paper.

*l.161/273: Is the Arctic Ocean included in the integral of the buoyancy flux?*

Yes, the Arctic Ocean is included in the integral of the buoyancy flux. We will add this explicitly in the revised paper.

*Fig.11: What is the averaging period shown here? What does the grey circle around the North pole indicate?*

What is shown is the mixed layer depth at the times corresponding to the CO2 concentrations indicated in the panel titles. There is no averaging in time implied in this figure. The grey circle around the North Pole is an artifact and will be removed.

*l.169-188: This part is difficult to read with the many "increases" and "decreases". Try to spell out more specifically whether the listed factors induce a buoyancy loss or gain. It can become difficult to correctly interpret increase and decrease if a property (such as M) can have different signs with small or large absolute values.*

Thanks for pointing this out. We agree and will try to make this section better readable by following the reviewer's suggestions.

*Fig.13 and related text: This figure is discussed very briefly, approximately with one and a half sentence. It might be worth to spend a few more words on this figure and to also make the connections between the left half and the right half clearer. Especially because the information in the right half has already been shown in Fig. 10 and 12. Also the relation between hosing and noisy freshwater forcing could be explained some more.*

Indeed, we agree that it makes sense to discuss the relation between the left and right part of this figure in some more detail and will do so in the revised paper.

*l.194-201: Compare diapycnal diffusivity results to previous work, e.g. diapycnal diffusivity seems to have played a key role in generating the DO oscillations under LGM conditions in Peltier&Vettoretti (2014).*

We will expand the discussion on the role of diapycnal diffusivity, including some previous work, as also suggested by Reviewer #2.

---

## Author Response (AR1)

**Reviewer #1**

*Willeit and colleagues present a large ensemble of CLIMBER-X simulations with various combinations of continental ice sheet configurations and atmospheric CO2 concentrations. This unprecedented ensemble allows them to analyse the physical conditions that determine the forcing range in which CLIMBER-X produces DO-like, millennial-scale climate variability. They find that this "sweet spot" is controlled by the sign of the surface buoyancy flux north of 55N. Millennial-scale transitions between week and strong AMOC states occur when the buoyancy flux north of 55N is about to switch sign. A strong/present day like AMOC occurs when the buoyancy flux is negative and deep water formation takes place in the Labador and Nordic Seas. When the buoyancy flux switches sign, this modern-like deep water formation pattern becomes unsustainable. The conditions under which this sign switch occurs are controlled by the boundary conditions. LGM-like ice sheets tend to enhance buoyancy loss, while low CO2 concentrations tend to decrease it. The balance of the two effects seems to be well captured by CLIMBER-X as the strongest DO-like variability occurs at realistic MIS3-like boundary conditions.*

*Some of the conclusions are not exactly new, e.g. the cancellation of the effects of ice sheet size and CO2 concentration. However the range and combination of covered boundary conditions is unprecedented, and the results are very relevant for the DO- and wider CP community and thus definitely worthy of publication. It is also very much appreciated that the authors define a metric that could be used to compare the physical conditions that control the "sweet spot" across models. Before publication, I would ask the authors to provide more context in some parts and to address a few issues as outlined in my comments below.*

We thank the reviewer for the positive appraisal of our work and the constructive comments.

*Major Comments*

*1. Introduction/Discussion: Please provide more context on what has already been suggested in terms of physical control on the sweet spot. At least Galbraith & de Lavergne (2019) and Klockmann et al (2018) provided some suggestions, e.g. the overall volume of Antarctic Bottom Water (AABW) present in the deep ocean, the density difference between AABW and North Atlantic Deep Water, presence of deep water formation in the Nordic Seas. Also spell out more directly how the additional CLIMBER-X simulations can help in pin-pointing the physical control across models. Because the physical control might also be model dependent.*

We have considerably expanded the introduction with a more extensive discussion of what controls AMOC strength in general, and what has been previously suggested in terms of control of the sweet spot in particular.
We have also substantially reworked the Appendix on the surface buoyancy, including a theoretical derivation of the *M* criterion, which outlines the rationale behind this measure and what assumptions go with it. We have also added the following text:
*'How effective M is in diagnosing convective instability will depend on the degree to which the assumptions and approximations in the derivation above are met, which could to some extent also be model dependent. Note that if under some circumstances the simplifying assumptions made in the derivation of the criterion in eq. D12 above are not met, a criterion for the stability of the convection can still be derived from the procedure above, but it will simply be a bit more complicated and will not strictly lead to the buoyancy criterion in eq. D14.'*

*2. I agree that the buoyancy flux analysis in this paper and the one in Klockmann et al (2018) cannot be compared directly one to one but at least a qualitative comparison should be possible and would actually strengthen the authors arguments even further. This could e.g. take place in the Discussion section.*

*Overall, the mode transitions in the experiments with PI ice sheets in Klockman et al also occur when net buoyancy flux over their NAtl&LabSea region changes from buoyancy loss to buoyancy gain (Klockmann et al use density instead of buoyancy, so the sign is flipped). In their Nordic Seas region, the buoyancy flux is close to zero for the $CO_2$ range where the transition takes place, so the Nordic Seas would not change the sign. This qualitative agreement makes the suggested metric M in the discussion of this manuscript even stronger.*

Calculating buoyancy over the deep water formation area(s), as in Klockman et al. (2018), is of limited use because this flux will be strongly negative as long as deep water formation continues. This is because most of the heat is released in this area, but only a small fraction of the freshwater flux enters the surface through this area (e.g. river runoff along continental margins). Therefore, this flux does not necessarily provide information about the stability of the AMOC. On the contrary, the sign of the buoyancy flux integrated over the entire Atlantic/Arctic ocean domain north of 55N, as shown in our paper, provides useful information about the (convective) stability of the AMOC and explains its instability under glacial conditions. In the revised paper we have added some further discussion on the rationale behind using integrated versus local buoyancy flux in diagnosing AMOC instability: *'The use of M to diagnose AMOC instability related to convection processes is based on the following idea. The northern North Atlantic and Arctic regions are characterized by a positive surface freshwater balance as a result of an excess of precipitation over evaporation in combination with freshwater input from rivers. The removal of this freshwater excess from the North Atlantic and Arctic regions can occur through (i) surface currents transporting low-salinity water to the south, (ii) sea ice export or (iii) deep mixing and evacuation of the freshwater through the deep ocean. In the case of the interstadial (DO) mode of the AMOC, the (iii) mechanism is the dominant one, while in the stadial mode the mechanisms (i) and (ii) dominate. As shown in Appendix D, the necessary condition for sustaining deep convection is a net negative surface buoyancy flux integrated over the whole northern North Atlantic and Arctic.'*

*One interesting difference can be seen in the effect of ice sheets on the thermal component. In Klockmann et al, the stronger net buoyancy loss with glacial ice sheets is due to increased heat loss over the deep convection sites, while in the present study, it is due to the reduced freshwater input. I do not have an immediate hypothesis where this difference might arise from. Perhaps it is simply due to the different areas of integration.*

This difference can indeed be at least partly explained by the different areas of integration (see Fig. 1 and Fig. 2 below). The changes in buoyancy due to glacial ice sheets are generally larger if considering the whole northern North Atlantic north of 55°N and Arctic (*M*) then if looking at the individual deep water formation regions as defined in Klockmann et al. (2018) (Fig. 1a and Fig. 2a). When looking at the North Atlantic and Labrador Sea region, also in CLIMBER-X the buoyancy decrease induced by ice sheets is explained by an increase in thermal buoyancy loss, similarly to Klockmann et al., (2018), at least for $CO_2$ concentrations between 220 and 280 ppm (Fig. 1b). In contrast, in CLIMBER-X the decrease in *M* with glacial ice sheets is largely caused by reductions in the net surface freshwater flux (Fig. 1c). Since most of the heat transported northward by the AMOC will be released over the

convection areas, the much stronger AMOC simulated with LGM ice sheets compared to present-day ice sheets (for a given CO2 concentration) will result in a strong increase in the surface sensible heat loss and consequent increase in buoyancy loss over the convection regions, which is well captured by the integration areas in Klockmann et al. 2018. However, the integration areas in Klockmann et al. 2018 capture only part of the changes in hydrological cycle and the resulting changes in the net surface freshwater fluxes.

That said, it is also clear from Fig. 15 that CLIMBER-X tends to show a larger haline buoyancy response between pre-industrial and LGM compared to PMIP models. This could be attributable to a substantial CLIMBER-X AMOC weakening at LGM, which results in a cooler northern North Atlantic and therefore a decrease in precipitation, while most PMIP models, and also the MPI-ESM used in Klockmann et al. 2018, show a strengthening of the AMOC at LGM.

We have added the following text to highlight this:

*'CLIMBER-X tends to show larger haline and thermal buoyancy responses between pre-industrial and LGM compared to most PMIP models. This could be attributable to a substantial CLIMBER-X AMOC weakening at LGM as opposed to most PMIP models, which acts to decrease the thermal buoyancy loss and decrease the haline buoyancy gain due to a cooler northern North Atlantic weakening the hydrological cycle.'*

[Figure]

*Figure 1 . Buoyancy flux integrated over different regions as a function of atmospheric CO2 concentration for simulations with different ice sheet configurations: the red lines are for experiments with interglacial ice sheets and the blue lines for full glacial (LGM) ice sheets . (a) Net buoyancy and (b) thermal and (c) haline buoyancy components. The solid lines are the integrated buoyancy measure M defined in the paper, while the dashed*

*lines represent the buoyancy integrated over the North Atlantic and Labrador Sea region as defined in Klockmann et al. (2018).*

[Figure]

*Figure 2 . Buoyancy flux integrated over different regions as a function of atmospheric CO2 concentration for simulations with different ice sheet configurations: the red lines are for experiments with interglacial ice sheets and the blue lines for full glacial (LGM) ice sheets. (a) Net buoyancy and (b) thermal and (c) haline buoyancy components. The solid lines are the integrated buoyancy measure M defined in the paper, while the dashed lines represent the buoyancy integrated over the Nordic Seas region as defined in Klockmann et al. (2018).*

*3. What is the role of sea ice in the buoyancy flux? Is the effect of freezing/brine release and melting included in the freshwater and heat budgets? Sea ice typically plays a big role in feedback loops regarding convection patterns. Even though it can be difficult to determine whether sea-ice is driving the change in the convection patterns or responding to it, it is still worth to be included more explicitly in the analysis.*

The effect of sea ice formation and melt is explicitly included in the computation of the surface buoyancy flux. This has been made more explicit in the revised paper. We also clarified under which circumstance it is important to account for the sea ice effect and when it can be neglected. As long as sea ice is formed and melted inside the area of integration of $M$ (north of 55°N), the net contribution of sea ice to the integrated buoyancy flux will be small and only due to the non-linear equation of state if sea ice is formed and melted in regions which differ in their sea surface temperature.

*4. It might be insightful to show the buoyancy flux also for the equilibrium simulations, e.g. in a similar style as Fig. 3 with buoyancy flux as the colour coding. That would help in linking the results from the transient and equilibrium simulations.*

Thanks for the suggestion. We have included a new figure (new Fig. 12) showing the average buoyancy $M$ as a function of atmospheric CO2 and ice sheet configuration for the equilibrium simulations.

**Minor Comments**

*l.3 "latitudinal reach" or "northward extent" instead of "latitude reach"?*

We changed it from 'latitude reach' to 'northward extent'.

*l.38-40: see major comment 1*

As outlined in the response to the major comment 1 above, we have substantially expanded this section in the revised paper.

*l.45: what is the climate-only setup? Are there other setups?*

CLIMBER-X also includes a global carbon cycle model and an ice sheet model, which are not used in the present study as CO2 and ice sheets are prescribed as constant over time. However, considering that this statement might confuse the readers, we have removed '*in a climate-only setup*'.

*l.75: How sensitive is the model to the area where the noise is applied? Why is it applied only locally and not globally?*

Initial sensitivity tests (not shown in the paper) indicated that the results are not very sensitive to the details of where noise is applied, as long as it covers the areas in the North Atlantic where deep water forms. Noise is introduced in the model to mimic synoptic-scale and interannual climate variability, and applying the same noise globally would be unrealistic as it would assume that this climate variability is globally uniform, which is definitely not the case. A more realistic, global, application of noise would require some kind of weather generator, which is beyond the scope of this study and would very likely have no impact on the results presented in the paper.
In the revised paper we have added the sentence: '*Sensitivity tests indicated that the model results are not very sensitive to the details of where the noise is applied, as long as it covers the areas in the North Atlantic where deep water forms*'.

*l.115: please also state the temperature changes over Greenland in the simulations and in the reconstructions. What does it mean if Greenland change is not capture well but the Iberian margin yes?*

In the revised paper we have explicitly added the modelled and reconstructed temperature changes in Greenland:
*"The amplitude of the simulated temperature variations over Greenland (~4°C) is underestimated compared to ice core reconstructions (~5-15°C)".*
The deficiency in the simulated Greenland temperature response in the model is somewhat

expected as the atmosphere in CLIMBER-X works best over relatively flat terrain, while the Greenland ice sheet is characterized by large slopes and the circulation over steep slopes is not properly resolved by the model. DO events are expected to affect mainly winter temperature in the northern North Atlantic, primarily as a response to the retreat in sea ice. This temperature changes are going to be largest in a relatively thin layer close to the surface and since in the atmosphere model the transport of heat is mostly horizontal, the warming over the ocean is not very efficiently transported to the summit of the Greenland ice sheet. Also other models, including many GCMs, tend to underestimate the DO warming over Greenland (e.g. Kuniyoshi et al., 2022; Malmierca-Vallet et al., 2024). We have added the following text to the revised paper:

*"The deficiency in the simulated temperature response over Greenland in the model is somewhat expected. DO events are expected to affect mainly winter temperature in the northern North Atlantic, primarily as a response to the retreat in sea ice. This temperature changes are going to be largest in a relatively thin layer close to the surface and since in the atmosphere model the transport of heat is mostly horizontal, the warming over the ocean is not very efficiently transported to the summit of the Greenland ice sheet. Also other models, including many GCMs, tend to underestimate the DO warming over Greenland* (e.g. Kuniyoshi et al., 2022; Malmierca-Vallet et al., 2024)*."*

*l.123-124: "The heat transport [...]" What do you base this sentence on? Is it based on previous studies (if yes, please cite)? Or do you infer it from your results (if yes, please elaborate shortly)?*

This sentence is based on our results, but from simulations not shown in the paper. We have therefore deleted this sentence in the revised paper.

*Fig.7: Please correct the caption. The interstadial sea-ice extent is drawn in dark teal and not grey*

Corrected, thanks.

*Fig.8: In the experiment description and in Fig.9 you mention a total of six noise amplitudes. Here you show only four. Why are 0.0625 and 0.125 not shown? Or did you not cover the full CO2 range for these amplitudes? If so please mention this in the experiment description.*

We performed the 0.0625 and 0.125 noise amplitude simulations only for a CO2 concentration of 170 ppm, which is why those noise levels are not included in Fig. 8. This has been clarified in the revised manuscript.

*Fig.9: Which CO2 concentration was used in the respective simulations displayed here?*

The simulations in the figure are for a CO2 concentration of 170 ppm. We have clarified this in the caption.

*l.152: Please briefly state, how do you define stable here (and elsewhere in the manuscript). Also, how realistic are the deep convection patterns in CLIMBER-X given the very coarse resolution?*

Here (and elsewhere), 'stable' has been removed as it did not add any relevant information. The coarse model resolution is obviously a limitation of our model. However, the present-day

deep convection patterns compare well to ocean reanalysis in the North Atlantic as shown in Fig. 13 in Willeit et al. 2022. There are no reconstructions of the mixed layer depth for DO Stadials and Interstadials, but some information on the convection patterns can be derived from sea ice extent reconstructions, which are tightly linked to the locations of deep water formation. As shown in Fig. 7 and discussed in the text, it seems that the CLIMBER-X sea ice extent change between Stadials and Interstadials is in qualitative agreement with reconstructions, providing some support for the simulated deep water formation patterns.

*l.154: "two modes" instead of "two stable modes". The "stable" in the latter half of the sentence ("are stable under the same CO2") is sufficient.*

Has been fixed, thanks.

*l.161/Fig.10: what about the smaller oscillations that occur around 160ppm with interglacial ice sheets and around 240ppm with mid-glacial ice sheets? In these cases, the buoyancy flux does not change sign.*

The smaller oscillations for interglacial ice sheets and CO2 around 160 ppm are not reflected in the buoyancy flux because they involve changes in convection pattern that are mostly confined to latitudes south of 55°N, which is therefore not reflected in $M$.
The oscillations at ~240 ppm for mid-glacial ice sheets involve a reorganization of deep water formation inside the domain north of 55°N. This therefore shows a clear imprint on $M$, but does not cause a change of sign of $M$, as convection remains present north of 55°N.
We have added this discussion in the revised paper:
*'The AMOC transition at CO2 ~160 ppm for interglacial ice sheets (Fig. 10a) is not reflected in M (Fig. 10b) because it involves changes in convection pattern that are mostly confined to latitudes south of 55°N. The sudden AMOC weakening at CO2 ~220 ppm for mid-glacial ice sheets (Fig. 10c) involves a reorganization of deep water formation inside the domain north of 55°N and therefore shows a clear imprint on M (Fig. 10d), but does not cause a change of sign of M, as convection remains present north of 55°N.'*

*l.161/273: Is the Arctic Ocean included in the integral of the buoyancy flux?*

Yes, the Arctic Ocean is included in the integral of the buoyancy flux. We added this explicitly in the revised paper and we also included a figure in the Appendix showing the area of integration.

*Fig.11: What is the averaging period shown here? What does the grey circle around the North pole indicate?*

What is shown is the mixed layer depth at the times corresponding to the CO2 concentrations indicated in the panel titles. There is no averaging in time implied in this figure. The grey circle around the North Pole was an artifact and has been removed.

*l.169-188: This part is difficult to read with the many "increases" and "decreases". Try to spell out more specifically whether the listed factors induce a buoyancy loss or gain. It can become difficult to correctly interpret increase and decrease if a property (such as M) can have different signs with small or large absolute values.*

Thanks for pointing this out. We agree and rephrased several sentences in this section in terms of buoyancy gain and loss.

*Fig.13 and related text: This figure is discussed very briefly, approximately with one and a half sentence. It might be worth to spend a few more words on this figure and to also make the connections between the left half and the right half clearer. Especially because the information in the right half has already been shown in Fig. 10 and 12. Also the relation between hosing and noisy freshwater forcing could be explained some more.*

In the revised paper we have added the following discussion of Fig. 13 (new Fig. 14):
*'Fig. 14 shows that a slow decrease in atmospheric $CO_2$ and a slow increase in freshwater forcing into the northern North Atlantic both produce a gradual decrease in buoyancy loss and eventually trigger an abrupt weakening of the AMOC when M switches from negative to positive. Convective instability can therefore also be triggered by directly perturbing the surface freshwater balance, which affects M (Fig. 14a,c). The noise that is applied to the surface freshwater flux in the model is thus also directly affecting the surface buoyancy flux and therefore facilitates the transition between different convection states. This also explains why larger noise amplitudes broaden the $CO_2$ range over which oscillations are observed in the model (Fig. 8).'*
Additionally, in the Conclusions and discussions section we have added the following:
*'In this paper, the effect of external freshwater forcing on AMOC stability has only been marginally explored, and a comprehensive analysis of AMOC stability in freshwater forcing and $CO_2$ phase space is presented in Willeit and Ganopolski (2024).*
*The changes in $CO_2$ concentration and ice sheet configuration applied in this study also strongly affect the hydrological cycle and thus the net freshwater flux in the Atlantic, but these changes have been taken into account by the model and treated as internal changes. At the same time, it should be noted that ice sheets are prescribed in all of our simulations, whereas in reality transient changes in ice volume over glacial-interglacial cycles will impact the freshwater balance of the northern North Atlantic and could have a pronounced effect on buoyancy and therefore the conditions favorable for the development of DO-like variability. Transient coupled climate - ice-sheet simulations will be required to address that.'*

*l.194-201: Compare diapycnal diffusivity results to previous work, e.g. diapycnal diffusivity seems to have played a key role in generating the DO oscillations under LGM conditions in Peltier&Vettoretti (2014).*

We have expand the discussion on the role of diapycnal diffusivity, including some previous work, as also suggested by Reviewer #2.
*'Previous work has explored the effect of ocean mixing on AMOC stability, with several studies showing that larger diapycnal mixing strengthens and stabilized the AMOC and increases the hysteresis width to freshwater forcing (e.g. Nof et al. (2007), Prange et al. (2003), Sijp and England (2006) and Schmittner and Weaver (2001)).*
*Peltier and Vettoretti (2014) and Peltier et al. (2020) discussed the role of different diapycnal diffusivities in shaping DO oscillations in their model and Malmierca-Vallet et al. (2023) note that the different representation of vertical mixing in climate models could explain why some models produce internal DO-like variability under specific boundary conditions, while others do not. In agreement with previous studies, larger diffusivities tend to make the AMOC stronger in CLIMBER-X,...'*

**Reviewer #2**

*Willeit et al. present and investigate DO-type millennial-scale oscillations from CLIMBER-X simulations. By analyzing North Atlantic surface ocean buoyancy fluxes, the authors provide further insight into the processes controlling convective stability and DO oscillations. The model indicates that transitions between different AMOC states occur when the buoyancy flux in the northern North Atlantic shifts from negative to positive, affecting convection patterns. Factors such ice sheet size, and CO2-induced cooling play crucial roles in stabilizing or destabilizing convection, shedding light on the mechanisms behind abrupt climate changes like DO events. The investigation of AMOC stability properties presented here is very comprehensive. In addition to the role of ice sheet size and CO2, the effects of climatic noise and ocean diapycnal mixing were also studied. The manuscript is well written, the results are very interesting and I enjoyed reading it very much. In my opinion, the study should be published in CP after the following points have been addressed.*

We thank the reviewer for the positive appraisal of our work and the valuable comments.

*- Previous studies have focused on the role of orbital parameters in DO-oscillations (e.g. Zhang et al. 2021; Kuniyoshi et al. 2022). Willeit et al. used present-day orbital parameters. How does this influence the results? I suggest to add a short discussion.*

For this study we have not performed a systematic analysis of the role of orbital forcing as we think that the combined effect of CO2 and ice sheets is sufficient to explain the concept of buoyancy control of DO events and adding a third dimension would result in unnecessary complications. However, for mid-glacial ice sheets we have run additional simulations showing that lower obliquity generally brings the system closer to the instability regime, resulting in DO-like oscillations being produced already for higher CO2 (Fig. 1 below). This is consistent with results presented by Zhang et al. (2021).
We plan to investigate the role of orbital forcing in more detail in future work using transient model simulations of the last glacial cycle.

[Figure]

Figure 1. Standard deviation of simulated AMOC time series for mid-glacial ice sheets and CO2 concentration ranging between 160 and 280 ppm for different obliquities. Precession and eccentricity are equal to present-day values.

*- CLIMBER-X underestimates the amplitude of Greenland temperature variations. Please discuss possible causes of this shortcoming.*

The deficiency in the simulated temperature response in the model is somewhat expected as the atmosphere in CLIMBER-X works best over relatively flat terrain, while the Greenland ice sheet is characterized by large slopes and the circulation over steep slopes is not properly resolved by the model. DO events are expected to affect mainly winter temperature in the northern North Atlantic, primarily as a response to the retreat in sea ice. This temperature changes are going to be largest in a relatively thin layer close to the surface and since in the atmosphere model the transport of heat is mostly horizontal, the warming over the ocean is not very efficiently transported to the summit of the Greenland ice sheet.
Also other models, including many GCMs, tend to underestimate the DO warming over Greenland (e.g. Kuniyoshi et al., 2022 and Malmierca-Vallet et al., 2024). We have added the following text to the revised paper:
*"The deficiency in the simulated temperature response over Greenland in the model is somewhat expected. DO events are expected to affect mainly winter temperature in the northern North Atlantic, primarily as a response to the retreat in sea ice. This temperature changes are going to be largest in a relatively thin layer close to the surface and since in the atmosphere model the transport of heat is mostly horizontal, the warming over the ocean is not very efficiently transported to the summit of the Greenland ice sheet. Also other models, including many GCMs, tend to underestimate the DO warming over Greenland* (e.g. Kuniyoshi et al., 2022; Malmierca-Vallet et al., 2024)*."*

*- The surface buoyancy flux analysis is very interesting. However, the authors do not explicitly consider the role of sea ice in controlling surface heat and freshwater fluxes. More discussion on sea ice effects would be necessary.*

The contribution of sea ice formation and sea ice melt to surface freshwater and heat fluxes is accounted for in the computation of the surface buoyancy flux. We have made this more explicit in the main text and also described it in more detail in the newly introduced theoretical derivation of the buoyancy criterion in the Appendix.

*- The authors describe an important role of the Laurentide ice sheet in "blocking part of the Pacific-to-Atlantic atmospheric moisture transport" (line 183). However, there should be additional effects of the ice sheet on moisture transports, e.g. through weakening of the hydrologic cycle by cooling the atmosphere. Please add further discussion to this topic.*

Following also the suggestions by the other Reviewers, we have extended the discussion of how the Laurentide ice sheet affects the surface freshwater balance in the model:
*'Prescribing LGM ice sheets leads to a decrease in the net Atlantic freshwater flux by ~0.1 Sv compared to experiments with present-day ice sheets, almost independently of the CO2 concentration, and is a result of the Laurentide ice sheet effectively blocking part of the Pacific-to-Atlantic atmospheric moisture transport in addition to the cooling induced by the presence of the ice sheets which weakens the hydrological cycle over the North Atlantic. Most of the reduction in freshwater flux occurs in the northern North Atlantic, in qualitative*

*agreement with Eisenman et al. (2009) and Sherriff-Tadano et al. (2021), thereby increasing the surface buoyancy loss in the deep-water formation regions.'*

*- The authors test the role of ocean diapycnal diffusivity and obtain interesting results. However, the discussion of the results comes up a little short here. Previous studies have explored effects of ocean mixing on AMOC stability. In particular, several studies showed that diapycnal mixing not only strengthens the AMOC but also enhances hysteresis width and the stability of the AMOC (e.g. Nof et al. 2007; Prange et al. 2003; Sijp and England 2006). I suggest to put the CLIMBER-X results into context considering previous work.*

As suggested by this Reviewer and Reviewer #1, we have expanded this section extending the discussion to include some previous work on the effect of diapycnal diffusion on AMOC strength and stability:
*'Extensive work has explored the effect of ocean mixing on AMOC stability, with several studies showing that larger diapycnal mixing strengthens and stabilizes the AMOC (e.g. Manabe and Stouffer, 1999; Ganopolski et al., 2001; Bryan, 1987; Nof et al., 2007; Prange et al., 2003; Schmittner and Weaver, 2001; Sijp and England, 2006). Peltier and Vettoretti (2014) and Peltier et al. (2020) discussed the role of different diapycnal diffusivities in shaping DO oscillations in their model and Malmierca-Vallet et al. (2023) note that the different representation of vertical mixing in climate models could explain why some models produce internal DO-like variability under specific boundary conditions, while others do not. In agreement with previous studies, larger diffusivities tend to make the AMOC stronger in CLIMBER-X,...'*

*- Line 144: "...which cannot be done with GCMs resolving synoptic processes". Yes, but in principle one could also add noise to the surface fluxes in GCMs. I suggest to rephrase to be more precise.*

We have rephrased this sentence to:
*'Our model has the advantage that it enables a separate investigation of the role of noise on DO dynamics, which can only be partly addressed with GCMs resolving synoptic processes, i.e. by adding additional noise on top of the internally generated variability. However, GCMs cannot remove the noise and can therefore not answer the question of whether noise is crucial for the existence of simulated DO-like events or not.'*

*- Figure 7: Add more information to the figure caption (i.e. which boundary conditions were used in this specific experiment?).*

The figure shows results from the simulation with mid-glacial ice sheets and a CO2 of 180 ppm. We have added this information to the caption.

*- Equation (D2) in line 266 describes the surface buoyancy flux. I am wondering whether the model uses a real freshwater flux formulation or virtual salt flux. Please clarify.*

The ocean model in CLIMBER-X is a rigid-lid model and we therefore use a virtual salt flux formulation for the surface freshwater flux, as described in detail in Willeit et al. 2022. We have clarified this in the revised manuscript.

**Reviewer #3**

*In this study, the authors explore the relation of surface buoyancy forcing and the initiation of the intrinsic oscillations (or threshold) of the AMOC in their earth system model. For this purpose, they conduct ensembles of simulations varying climatic forcing and atmospheric noise in the model. The model reproduces the modern and the LGM AMOC reasonably well. Also it reproduces the occurrence of the intrinsic oscillation of the AMOC under mid-glacial boundary conditions. The sets of experiments with different magnitudes of noise show the effect of noise in increasing the window of opportunity to cause the AMOC variability. Lastly, the authors explore the role of integrated buoyancy forcing (M) in predicting the initiation of the AMOC variability. Particularly, they show that the threshold type behavior of AMOC occurs when "M" approaches to zero.*

*I'd like to thank the authors for their effort in running so many exciting simulations. Especially, I find the experiments with different magnitude of noise very exciting since it is technically challenging to do so in AOGCMs! Furthermore, the authors are investigating an important question, "What controls the condition of the sweet spot?/Why DO cycles occur frequently under mid-glacial periods", which is of highly interest to the readers of Climate of the Past. Therefore, I think these results should be published. However, while the presented figures are exciting, I feel that this study has lots of rooms for improvements in the writing part. For example, in the paper, "M" is introduced in a heuristic way, but the physical reasoning of why M can be a good index is not fully discussed and it is not also compared to existing literatures. Therefore I would recommend major revision. Below summarizes my criticism.*

*Best wishes,*

*Sam Sherriff-Tadano*

We would like to thank the reviewer for the valuable comments on our paper, which helped to clarify some aspects of our presented research in the revised version of the manuscript.

**General Comments**

*1. Why focus only over the North Atlantic?*

*The authors focuses on the role of buoyancy forcing over the North Atlantic building on their previous work (Ganopolski and Rahmstrof 2001). While I agree that the North Atlantic is a very important region, I'm aware that there are quite a few other studies claiming the importance of the buoyancy forcing or density over the Southern Ocean in controlling the glacial AMOC (Buizert and Schmitner 2015, Sun et al. 2020, Oka et al. 2021). Perhaps, for this particular model, the North Atlantic could be the most critical regions, but there are other modeling studies suggesting the importance of both North Atlantic and Southern Ocean. For example, Sun et al. (2020) showed the importance of density contrast between NADW and AABW, rather than the buoyancy flux itself, in controlling the glacial (LGM) AMOC. I think it would be reasonable to point out these previous studies and then explain why this study focuses only on the buoyancy flux over the North Atlantic.*

We agree that the AMOC strength is controlled by numerous factors and we have expanded the introduction to discuss this in some more detail, following also the suggestions by the

reviewer. However, the main purpose of our paper is not to try to explain the strength of the AMOC, but to propose a measure to diagnose convective instability and the associated abrupt AMOC changes. We propose that the integrated buoyancy flux over the northern North Atlantic is a valid measure of that and consequently focus on the surface buoyancy flux over this region. In Appendix D we now also provide a theoretical derivation of the integrated buoyancy criterion, which will hopefully help to further clarify why we focus on the northern North Atlantic region. Of course, the buoyancy flux over the northern North Atlantic and its dependence on boundary conditions does also depend on the AMOC state in general and its meridional heat transport in particular, which is at least partly controlled by processes acting outside of this particular region.

In the revised paper we have extended the introduction with:

*'It is well known that the AMOC is controlled by many factors (e.g. Kuhlbrodt et al., 2007; Nayak et al., 2024), including wind stress, surface buoyancy fluxes and diapycnal mixing. Several studies also suggest the importance of the Southern Ocean and Antarctic bottom water formation in controlling the strength and depth of the AMOC under different climate conditions (e.g. Sun et al., 2020; Oka et al., 2021; Buizert and Schmittner, 2015). However, in this work our main aim is not to explain what controls the strength of the AMOC, but rather focus on what leads to AMOC instability under glacial conditions. Clarifying this is important in order to understand why in reality the DO events occurred under a broad range of glacial climate and boundary conditions, but not during interglacials and peak glacial conditions (Barker et al., 2011; Hodell et al., 2023). It has been suggested that the appearance of DO events is controlled by $CO_2$ (Vettoretti et al., 2022), the size of the Northern Hemisphere ice sheets (Zhang et al., 2014; Klockmann et al., 2018; Brown and Galbraith, 2016), and orbital configuration (Zhang et al., 2021). Malmierca-Vallet et al. (2024) recently highlighted the possible role of $CO_2$ concentration in explaining DO variability across different models, independently of the size of the Northern Hemisphere ice sheets and other boundary conditions. However, while the concept of a 'sweet spot' for the occurrence of DO-like variability has recently gained considerable attention, what physical conditions control where it is located in the ice sheet–$CO_2$–orbit space in the different models has remained largely unexplained. Klockmann et al. (2018) and Galbraith and de Lavergne (2019) suggested that the AMOC instability is controlled by the surface density difference between the Southern Ocean and the North Atlantic deep water formation sites, with low $CO_2$, low obliquity and relatively small ice sheets favoring a weak AMOC that is closer to instability. A critical role of Arctic sea ice has also been suggested (Loving et al. 2005) as well as changes in surface winds by glacial ice sheets (Sherriff-Tadano et al. 2021a). Here we use a large number of simulations with an Earth system model to explore the physical control mechanisms behind DO-like variability and propose a key role of the surface buoyancy flux over the northern North Atlantic in controlling convective instability and the associated abrupt changes in the AMOC.'*

*2. Comparison with previous studies*

*I appreciate the authors effort in shorting the Introduction and Discussion, however I think the authors are missing important previous studies that tried to answer similar scientific question "What controls the condition of the sweet spot?/Why DO cycles occur frequently under mid-glacial periods". For instance, previous studies have pointed out the potential importance of Antarctic temperatures (Buizert and Schmittner, 2015; Kawamura et al., 2017), Arctic sea ice (Loving and Vallis, 2005) or changes in surface winds by glacial ice sheets (Sherriff-Tadano et al., 2021a) in initiating the DO-like climate variability frequently during the mid-glacial period. None of the above mentioned studies have explored the role of*

*integrated buoyancy flux over the North Atlantic, but I feel it is beneficial to describe these studies so that the readers can learn some of the history of this research topic.*

As outlined in the response to comment 1 above, we have substantially expanded the Introduction section, providing a bit more background information on previous studies.

*3. Why is it better to integrate the buoyancy flux over the entire northern North Atlantic?*

*Here, I'm concerned about the role of sea ice as some of the other reviewers. Previous studies showed the importance of sea ice transport through the Denmark Strait and its melting over the NADW formation region in weakening the AMOC (Born et al. 2010, Vettoretti and Peltier 2018). However, when the buoyancy flux is integrated over the entire region, the spatial heterogeneity in the sea ice-related freshwater flux will be removed.*

There is no doubt that sea ice is important for the AMOC in general and for DO events in particular. However, there is no reason to believe that any specific spatial heterogeneity of sea ice is needed to explain DO events. A lot of attention (probably too much) is now being paid to a "sweet spot" problem. However, DO events are a very persistent feature of the glacial world, and while the first DO events during the last glacial cycles (DO25) occurred under climate conditions which were very close to interglacial conditions, the DO2 event occurred just prior to the LGM. Obviously, sea ice properties (thickness, extent, transport) were very different during different DO events and it therefore is extremely unlikely that 30-meter thick Arctic sea ice (as simulated by Vettoretti and Peltier, 2018) is needed to explain real DO events. We are not denying the complexity of the real world, but we are asking a very reasonable question: why was the glacial AMOC fundamentally unstable? And by using the metric $M$, we propose an answer to this question.

*Under this condition, it is not straightforward why M can be a good predictor for the initiation of sweet spot/threshold.*

Unfortunately, in the first version of the manuscript we only postulated this metric $M$ and showed that it works in our model, but we did not explain the physical reasoning behind this concept. We have now corrected this oversight and presented the scientific justification for this metric in Appendix D, together with a discussion of its possible limitations.

*Perhaps in this model, I speculate that following two points could be important; 1. Sea ice forms and melts at the same region, hence the sea ice-related freshwater flux is not so important in the first place, or 2. The regional contrast in salinity induced by sea ice formation and melting is removed by advection of salt by oceanic currents.*

The correctness of the first assumption depends on the choice of the "region". In our case (the ocean north of 55°N) a certain amount of sea ice is transported away from the region, but this is automatically accounted for in the metric M, which also includes fluxes from sea ice formation and melt, as stated more clearly in Appendix D now. The second assumption is incorrect because our model does simulate significant contrast in salinity.

*Please discuss why it is better to integrate the buoyancy flux over the northern North Atlantic, rather than focusing over the NADW formation region.*

Calculating buoyancy over the deep water formation area(s) is of limited use because this flux will be strongly negative until deep water formation continues. This is because most of the heat is released in this area, but only a small fraction of the freshwater flux enters this area through the surface. Therefore, this flux does not provide information about the stability of the AMOC. On the contrary, the buoyancy flux integrated over the entire ocean domain north of 55N, as shown in our paper, provides useful information about the (convective) stability of the AMOC and explains its instability under glacial conditions. Of course, the accuracy of this metric depends on several assumptions, in particular that the southward export of freshwater in the upper ocean through 55N is small compared to the total freshwater balance of the Arctic and North Atlantic north of 55N.

The effect of sea ice formation and melt is explicitly included in the computation of the surface buoyancy flux. However, as long as sea ice is formed and melted inside the area of integration of $M$ (north of 55°N), the net contribution of sea ice to the integrated buoyancy flux will be small and only due to the non-linear equation of state if sea ice is formed and melted in regions which differ in their sea surface temperature.

In the revised paper we have added some further discussion on the rationale behind using integrated versus local buoyancy flux in diagnosing AMOC instability:

*'The use of M to diagnose AMOC instability related to convection processes is based on the following idea. The northern North Atlantic and Arctic regions are characterized by a positive surface freshwater balance as a result of an excess of precipitation over evaporation in combination with freshwater input from rivers. The removal of this freshwater excess from the North Atlantic and Arctic regions can occur through (i) surface currents transporting low-salinity water to the south, (ii) sea ice export or (iii) deep mixing and evacuation of the freshwater through the deep ocean. In the case of the interstadial (DO) mode of the AMOC, the (iii) mechanism is the dominant one, while in the stadial mode the mechanisms (i) and (ii) dominate. As shown in Appendix D, the necessary condition for sustaining deep convection is a net negative surface buoyancy flux integrated over the whole northern North Atlantic and Arctic.'*

*4. Bit more discussion on the role of noise?*

*Fig. 8d and f reminded me of different characteristics of intrinsic oscillations obtained from CESM(Vettoretti et al. 2022)/MIROC(Kuniyoshi et al. 2022) and MPI (Klockmann et al. 2018). This could be very speculative, but if the authors agree, it might be interesting to point out the potential role of noise in causing different shapes of AMOC variability among models.*

This is actually a good point, thanks. We have included the following sentence mentioning the possible role of noise in explaining the different AMOC response in different models:
*'Larger noise levels produce oscillations that are more symmetric and with a shorter period (compare Fig. 9d and f) and could possibly to some extent explain the different characteristics of intrinsic oscillations obtained by Vettoretti et al. (2022) and Kuniyoshi et al. (2022) as opposed to Klockmann et al. (2018).'*

**Specific Comments**

*L51: Please describe the climate sensitivity of the model here since it is one of the fundamental metric.*

The climate sensitivity of the model is ~3.1°C. We have added this information to the revised paper.

*L96-97: Would be suitable to cite Eisenman et al. (2009) and Sherriff-Tadano et al. (2021b) since they discuss the role of changes in atmospheric freshwater flux by ice sheets in intensifying the AMOC.*

We added a reference to these papers in line 184, where the effect of ice sheets on surface freshwater forcing is discussed.

*L209-215: Fig. 10a and b show a threshold type behaviour of AMOC around 160ppm of $CO_2$ even when the value of M is negative. Is this related to the miss-choice of $\varphi_M$? If so, it would be worth discussing it here.*

The oscillations for interglacial ice sheets and CO2 around 160 ppm are not reflected in the buoyancy flux because they involve changes in convection pattern that are mostly confined to latitudes south of 55°N, which is therefore not reflected in M. We have added this to the revised paper:
*'The AMOC transition at CO2 ~160 ppm for interglacial ice sheets (Fig. 10a) is not reflected in M (Fig. 10b) because it involves changes in convection pattern that are mostly confined to latitudes south of 55°N'*

---

## Author Response (AR2)

**Response to Reviewer #2**

We would like to thank again the reviewer for his second round of comments on our paper and provide a response to the comments in blue below.

**Summary**

This is my second review of the paper by Willeit and others. In the revised manuscript, the authors have expanded the Introduction by adding relevant previous studies exploring the potential causes of the occurrence of DO cycles at mid glacial period. The authors have also included the theoretical background of the new index M in the appendix and added a new discussion on the potential role of the noise in causing different shapes of DO-like variabilities simulated by different coupled models.

I think the authors have done a nice work in addressing my concerns that I raised in the first review. While I have some small comments and suggestions, I think that the M can be a good index to inform the possibility of the AMOC instability for other models, which are trying to simulate intrinsic variations in the AMOC similar to DO cycles. I would like to thank the authors for their effort in the revision and would be happy to recommend the manuscript for publication after minor corrections.

**Minor comments**

1. Implication of the areal integration of the buoyancy flux
The impact of sea ice transport and the related salinity changes within the North Atlantic-Arctic domain on the M and the stability of AMOC is still a bit vague to me. For example, Born et al. (2010) showed that a decrease in net radiative forcing over the Arctic modified the transports of sea ice and salinity within the domain and caused a weakening of the ocean convection and AMOC. In this case, although the net Fw within the domain is zero, AMOC weakened and approached destabilization. Of course, I know the authors do not focus on the strength of AMOC, but as shown in Figure 10, there is a strong relationship between AMOC strength and M, making it difficult to separate the two. Therefore, I am concerned that sea ice transport within the domain may alter the relationship between M and AMOC (for example, even if M is non-zero and has a negative value, AMOC could still destabilize depending on the sea ice and salt transport within the domain). Obviously, the magnitude of this effect should largely depend on models, and as the authors suggest, in the case of the 35m sea ice simulated in Vettoretti's model, the impact could be quite substantial, while in other models, the impact might be smaller. However, I feel that this is a good point to make in the discussion. Just adding some words or a sentence on L286-287 or wherever appropriate would be sufficient.

In our paper we discuss extensively the strong interaction between AMOC and sea ice, although the most important aspect of this interaction is the control of sea ice extent by the northward heat transport of the AMOC. We do not fully understand why the reviewer thinks that sea ice transport (which is of course included in our model) could invalidate the relationship between the integrated buoyancy flux *M* and the stability of deep water formation in the North Atlantic. Actually, one of the main strengths of our "theory" is that we suggest that redistributions of freshwater within the domain north of 55°N are

irrelevant, and this includes sea ice transport.

The reviewer cites the work of Born et al. (2010) as an example. However, what Born et al. found is that under the Eemian (126 ka) orbital configuration, a new deep convection site (in addition to two others) appeared south of Greenland (and south of 55°N, i.e. outside our domain) that does not exist under pre-industrial and 116 ka conditions. Born et al. explained the appearance of this convection site by a reduction in sea ice transport through the Denmark Strait at 126 ka compared to 116 ka. However, the changes in convection locations in their study occurred for changes in orbital configuration, that probably have significant effects on buoyancy through changes in surface temperature, AMOC heat transport and possibly also the precipitation-evaporation balance, in addition to sea ice transport changes.

In any case, we have clearly outlined the assumptions entering the derivation of our stability criterion in the main text and in the Appendix, and our "theory" can only be challenged by explicitly calculating $M$ with other models and demonstrating that in other models this criterion does not explain the convective (in)stability of the AMOC.

To make this clearer we added the following sentence to the discussion section:
*'Ultimately, estimations of M and its relation with convection and AMOC in other models will be needed to assess the robustness of our instability criterion.'*

L185: I had a difficulty understanding this sentence. Could you rephrase it?

We have rephrased it to:
*'Under some conditions, e.g. for present day ice sheets and CO2 between 220 and 240 ppm, two modes of the AMOC corresponding to different convective patterns are stable for the same CO2 (Fig. 10a), but this is not a pre-requisite for the occurrence of internal oscillations.'*

L377-385: Could you add a bit more explanation of why $\Psi_2 (S_*-S_2)$ could be small and neglected? In the same paragraph, it says S2 is significantly lower than S*, which means that (S*-S2) could be quite large.

The value S2 in our conceptual model is indeed "significantly lower" than 35 psu (it is at present about 33 psu). However, what is crucial for our theory is whether the following assumption is valid under glacial conditions: "2) Most of the net freshwater flux entering the North Atlantic/Arctic domain is mixed downward in the deep convection areas and then transported away by oceanic currents below the surface layer". Since under glacial conditions both the Bering Strait and the passage between Canada and Greenland were closed, and the only escape route was through the Denmark Strait, we believe that this assumption is a reasonable one and this is confirmed by the results of our model simulations.
We have added the following to make this clearer:
*' In our model this flux is small (generally <0.01 Sv) and can be neglected'.*

Fig. 12: Love this figure!